# Primal-Spectral Generative Modeling: Fast Analytical Generation via Pseudoinverse Lévy Inversion

Zhiyuan Ouyang [* 1 3]   Simei Huang [* 4]   Zuokai Wen [* 2]   Xiangyun Zhang [1]   Junchi Yan [2 3]

## Abstract

A probability distribution $\mathbb{P}$ is a measure on a $\sigma$-algebra, assigning mass to sets rather than points, which poses a challenge for the training of neural networks that often struggle to reconstruct the global topology of continuous manifolds from sparse samples. We mitigate this issue by transforming $\mathbb{P}$ into a continuous function via spectral methods, providing theoretical guarantees for the convergence of the learned distribution to the true distribution. Specifically, we introduce a network, PriSpecNet, with a single-function evaluation (1-NFE) Pseudoinverse Lévy Inversion (PiLI) solver that regards generation as a fast analytical problem, eliminating the need for iterative numerical integration while maintaining full compatibility with the stochastic interpolants. We test our PriSpecNet in two applications: for time series, it unifies generation and forecasting, outperforming state-of-the-art (SOTA) baselines with Context-FID reductions of 50.0%, 41.5%, 80.6%, and 63.1% on Sines, Solar, ETTh, and Stock benchmarks, respectively, also decreasing forecasting MSE by 29.8% on Solar and 23.8% on Stock. For ImageNet $256 \times 256$, 1-NFE PiLI achieves a competitive FID of **1.66** using only **26** Gflops, representing a **170×** reduction in total Gflops compared to the 4,436 Gflops required by the 25-NFE DPM-Solver++.

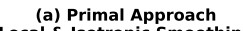

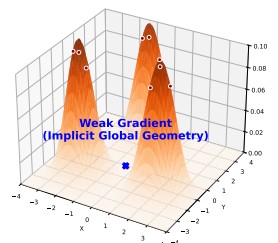 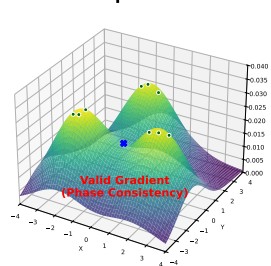

*Figure 1.* **Gradient landscape comparison for noisy data $x_t$ representations.** The horizontal axes $(X, Y)$ denote the 2D data space $\mathbb{R}^2$, while the vertical axis denotes the magnitude of the probability density $p_t$ or the spectral surface $\phi_{x_t}(u)$. (a) Primal Domain: The $x_t$ distribution forms a sparse empirical distribution $P_t = \frac{1}{N} \sum \delta(x_t^{(i)})$. Local smoothing results in disconnected modes and vanishing gradients $\nabla_{x_t} \log p_t(x_t) \approx 0$ at low-density regions. (b) Spectral Domain: Transform the $x_t$ into the global CF $\phi_{x_t}(u) = \mathbb{E}[e^{iu^\top x_t}]$ yields persistent gradients $\nabla_{x_t} \phi_{x_t}(u) = iu \odot \phi_{x_t}(u)$. This analytic form preserves phase consistency across the continuous surface, ensuring valid guidance throughout the domain.

## 1. Introduction

Generative models have experienced fast advancement, evolving from elementary distribution matching to capturing high-dim manifolds, which enables their expansion beyond theoretical benchmarks into practical, complex applications. Remarkable efficacy has been shown across a spectrum of domains (Ali et al., 2025), e.g. high-resolution image synthesis (Esser et al., 2024), video synthesis (Bar-Tal et al., 2024; Wan et al., 2025), and unified speech generation (Mehta et al., 2024; Chen et al., 2025b). Beyond perceptual modalities, these frameworks have extended to discrete text modeling (Lou et al., 2024; Ansari et al., 2024; Zhu et al., 2025), time series forecasting (Gao et al., 2025; Kong et al., 2020; Yuan & Qiao, 2024; Kollovieh et al., 2024), and scientific applications like biomolecular structure prediction (Abramson et al., 2024; Wohlwend et al., 2025).

Continuous Normalizing Flows (Chen et al., 2018) established Probability Flow Ordinary Differential Equation (PF-ODE), grounding Flow Matching (Lipman et al., 2022; Liu et al., 2022). Mean Flow (Geng et al., 2025a) and iMF (Geng

---

[*]Equal contribution   [1]School of Mathematical Sciences, Key Laboratory of MEA & Shanghai Key Laboratory of PMMP, East China Normal University, Shanghai, China [2]School of Artificial Intelligence, Shanghai Jiao Tong University, Shanghai, China [3]Shanghai Innovation Institute, Shanghai, China [4]Guilin Institute of Information Technology, School of Business, Guilin, China. Correspondence to: Xiangyun Zhang <xyzhang@math.ecnu.edu.cn>, Junchi Yan <yanjunchi@sjtu.edu.cnu>.

*Proceedings of the 43rd International Conference on Machine Learning*, Seoul, South Korea. PMLR 306, 2026. Copyright 2026 by the author(s).

*Table 1.* **Methodological comparison of generative paradigms.** While existing methods rely on iterative solvers or optimization, our method analyzes the CF within the framework of Distribution Interpolation (Albergo et al., 2025), enabling Analytical 1-NFE Generation via Lévy's Continuity Theorem (Ash & Doléans-Dade, 2000). $\hat{s}/\hat{v}$ denotes learned score/velocity, $\phi$ the CF, and $\mathscr{L}_{\text{IPF/Match}}$ minimizes the divergence from the entropy-regularized optimal transport path via iterative marginal fitting or direct drift regression.

| Paradigm | Principle | Training Objective | Generation Method | Literature |
|---|---|---|---|---|
| ***I. Primal Generative Approaches*** | | | | |
| Diffusion Models | Stochastic Calculus | $\mathbb{E}[\|\hat{s}-s\|_2^2]$ | | (Ho et al., 2020; Song et al., 2020) |
| Flow Matching | Optimal Transport | $\mathbb{E}[\|\hat{v}-v\|_2^2]$ | Iterative Solvers | (Lipman et al., 2022; Liu et al., 2022) |
| Schrödinger Bridge | Entropic Optimal Transport | $\mathscr{L}_{\text{IPF}}$ / $\mathscr{L}_{\text{Matching}}$ | (SDE/ODE) | (Tang et al., 2024) |
| Stochastic Interp. | Distribution Interpolation | $\mathbb{E}[\|\hat{v}-v\|_2^2]/\mathbb{E}[\|\hat{s}-s\|_2^2]$ | | (Albergo et al., 2025) |
| Improved Mean Flow | Straight-Path Interpolation | $\mathbb{E}[\|\hat{v}-v\|_2^2]$ | 1-Step Generator | (Geng et al., 2025b) |
| ***II. Spectral Generative Methods*** | | | | |
| Spectral Learning | CF / Moment Matching | $\|\phi-\hat{\phi}\|_2^2$ | Iterative Optimization | (Li et al., 2015; Ansari et al., 2020) |
| ***III. Primal-Spectral Generative Methods*** | | | | |
| **PriSpecNet (ours)** | Lévy's Thm & Dist. Interp. | $\mathbb{E}[\|\phi-\hat{\phi}\|_2^2]+\mathbb{E}[\|\hat{v}-v\|_2^2]$ | **Pseudoinverse Lévy Inversion** | (Ash & Doléans-Dade, 2000) |

et al., 2025b) optimized this via straight trajectories for efficient one-step generation. Unified with Diffusion Models (Ho et al., 2020; Song et al., 2020) under Stochastic Interpolants (Albergo et al., 2025), we define a process $x_t$ on $(\Omega, \mathscr{F}, \mathbb{P})$ connecting prior $x_1 \sim P_1$ to data $x_0 \sim P_0$. Since direct modeling of $P_t$ is intractable (Fournier & Guillin, 2015), neural approximations rely on the sparse empirical measure $P_0 = \frac{1}{N}\sum \delta(x_0^{(i)})$, regressing a vector field $v_t(x)$ solving the PF-ODE. This sparsity challenges manifold learning. While standard Gaussian smoothing provides gradients, it is inherently *local* and isotropic, often failing to capture *global* topological structures or multimodal dependencies. In contrast, our spectral approach transforms the distribution into the Characteristic Function (CF), a continuous global representation. This encodes both local and global structures, preserving phase consistency even under sparse sampling to provide valid gradients that guide the model toward the correct global topology, as shown in Fig. 1.

While Latent Generative Models (Rombach et al., 2022) and tokenizers Repa-E (Leng et al., 2025) or VQ-VAEs (Razavi et al., 2019) improve efficiency, their compression acts as a low-pass filter, compromising high-frequency details. We propose a primal-spectral framework to achieve latent-like efficiency directly in pixel space without compression. By transforming distributions into CFs to strictly bound Fisher divergence, we bridge the sparse primal and spectral duals, unifying training with fast deterministic sampling. **The contributions are:**

**1) Theoretical Framework for Spectral Training:** We provide a theoretical analysis of spectral training, proving that minimizing the spectral density difference effectively squeezes the Fisher divergence bounds. Based on this, we derive a tractable and numerically stable spectral loss.

**2) Efficient Primal-Spectral Modeling:** We propose effi-

cient Primal-Spectral Modeling, implemented via the tailored PriSpecNet in Fig. 2, and show that this architecture accelerates training convergence and inference speed in pixel space, overcoming the spectral bottlenecks and information loss typical of Latent Generative Models.

**3) Advanced Time Series Generative Tooling:** It establishes a new SOTA, significantly outperforming Auto-Regressive Moving Diffusion Models (ARMD) (Gao et al., 2025). On the ETTh dataset, we achieve an 80.6% reduction in Context-FID (lowering it from 0.093 to 0.018). Similarly, for forecasting tasks, we decrease the MSE by 29.8% on the Solar dataset and 23.8% on Stock, demonstrating robust consistency across both generation and prediction tasks.

**4) 1-NFE Pseudoinverse Lévy Inversion:** We propose PiLI, a deterministic solver that leverages spectral reconstruction to yield a closed-form solution. It enables true 1-NFE generation, delivering an approximate **170×** efficiency improvement in computational cost, slashing Gflops from 4,436 to just 26 compared to DPM-Solver++ (Lu et al., 2025) while securing a FID of 1.66 on ImageNet-256.

## 2. Related Work

**Primal Generative Frameworks.** Modern generative modeling is dominated by frameworks operating directly in the primal Euclidean space. Continuous Normalizing Flows (Chen et al., 2018) established the foundational principles of ODE-based probability paths, directly paving the way for Flow Matching (Lipman et al., 2022; Liu et al., 2022). Furthermore, Schrödinger Bridges (Tang et al., 2024) address the entropy-regularized optimal transport problem, enabling finite-time transitions between arbitrary distributions via iterative proportional fitting. Most recently, the theory of Stochastic Interpolants (Albergo et al., 2025) has

unified these approaches with Diffusion Models (Ho et al., 2020; Song et al., 2020), formulating generation as regressing a time-dependent velocity field that transports a prior $P_1$ to the data $P_0$. Within this unified landscape, iMF (Geng et al., 2025b) further optimizes the transport dynamics by enforcing straight-path interpolation, thereby achieving efficient high-fidelity generation in a single step. However, despite these advancements, these primal methods fundamentally rely on interacting with the empirical distribution, often struggling to capture global topological structures from sparse samples.

**Spectral Generative Models.** Beyond the primal domain, spectral learning methods offer a functional perspective by modeling the distribution's spectral statistics. Spectral Generative Models, such as Generative Moment Matching Networks (Li et al., 2015; Gretton et al., 2012) and CF matching approaches (Ansari et al., 2020), aim to minimize the discrepancy between the CFs of the generated and real distributions. Related frequency-domain generative studies have also shown the value of dual-space supervision through Fourier-space perceptual losses (Fuoli et al., 2021), Fourier-enhanced adversarial synthesis (Zhao et al., 2022; Zhang et al., 2025b), function-space adversarial learning (Adler & Lunz, 2018), and sine-cosine dual analyses of GAN dynamics (Asokan & Seelamantula, 2023). Alias-free generative architectures further reinforce this view by revealing how spatial transformations correspond to structured phase behavior in the spectral domain (Karras et al., 2021). While these methods successfully leverage global spectral information for training, they lack an explicit inversion mechanism. Consequently, generation typically relies on iteratively optimizing samples to match spectral statistics, which faces the ill-posed pre-image problem and lacks the tractability of direct inversion.

**Primal-Spectral Generative Modeling.** As summarized in Tab. 1, a fundamental distinction of our framework lies in its generative mechanism. Unlike primal methods that approximate vector fields for iterative integration, and unlike existing CF-based models that rely on black-box optimization, we target the **spectral of the probability distribution** with a mathematically grounded inverse. This choice bridges the gap between theory and practice: 1) It resolves the bottleneck of existing spectral models by providing a direct Analytical Inverse PiLI for the CF; 2) Grounded in Lévy's Continuity Theorem (Ash & Doléans-Dade, 2000), minimizing the spectral error guarantees weak convergence to the true distribution, offering a rigorous alternative to unstable moment-matching paradigms.

## 3. Preliminary

**Forward Interpolant Framework.** We formulate the generative task under the framework of Stochastic Inter-

polants (Albergo et al., 2025), a unified perspective for constructing probability paths between distributions. We define a stochastic interpolant $\{x_t\}_{t \in [0,1]}$ that interpolates between a data distribution $P_0$ and a prior distribution $P_1$. This interpolant is explicitly constructed via a linear combination of data sample and noise variable:

$$x_t = s(t)x_0 + \sigma(t)x_1, \qquad (1)$$

where $x_1 \sim \mathcal{N}(0, I_d)$ is standard Gaussian noise. The functions $s(t)$ and $\sigma(t)$ denote differentiable scaling and noise schedules. It has been established that the marginal densities of this interpolant satisfy the Itô SDE on a filtered probability space $(\Omega, \mathcal{F}, \{\mathcal{F}_t\}_{t \in [0,1]}, \mathbb{P})$:

$$\mathrm{d}x_t = \frac{\dot{s}(t)}{s(t)}x_t \, \mathrm{d}t + \sqrt{2\sigma(t)\dot{\sigma}(t) - 2\frac{\dot{s}(t)}{s(t)}\sigma(t)^2} \, \mathrm{d}W_t, \quad (2)$$

where $\{W_t\}_{t \geq 0}$ is an $\mathcal{F}_t$-adapted standard Wiener process. The interpolant coefficient here is specifically derived to preserve the marginal variance structure defined by $\sigma(t)^2$.

To realize generation, we consider the time-reversal of this stochastic process, evolving from prior $x_1$ to data $x_0$, as governed by the reverse-time SDE (Anderson, 1982):

$$\mathrm{d}x_t = \left[ \frac{\dot{s}(t)}{s(t)}x_t - \left( 2\sigma(t)\dot{\sigma}(t) - 2\frac{\dot{s}(t)}{s(t)}\sigma(t)^2 \right) \nabla_{x_t} \log p(x_t) \right] \mathrm{d}t$$
$$+ \sqrt{2\sigma(t)\dot{\sigma}(t) - 2\frac{\dot{s}(t)}{s(t)}\sigma(t)^2} \, \mathrm{d}\widetilde{W}_t, \qquad (3)$$

where $\{\widetilde{W}_t\}_{t \in [0,1]}$ denotes the corresponding reverse-time Wiener process.

**Deterministic Reverse Process.** Alternatively, the evolution of the marginal densities $P_t$ follows the deterministic PF-ODE. Expressing the drift via interpolant coefficients $s(t)$ and $\sigma(t)$ yields the particle flow trajectory:

$$\mathrm{d}x_t = \underbrace{\left[ \frac{\dot{s}(t)}{s(t)}x_t - \left( \sigma(t)\dot{\sigma}(t) - \frac{\dot{s}(t)}{s(t)}\sigma(t)^2 \right) \nabla_{x_t} \log p(x_t) \right]}_{v(x_t)} \mathrm{d}t,$$
$$(4)$$

where the bracketed term defines the time-dependent velocity field $v(x_t)$. Instead of indirect score estimation, we directly parameterize this field via a neural network $\hat{v}(x_t)$. By adopting the linear interpolation schedule $s(t) = 1 - t$ and $\sigma(t) = t$, the target velocity simplifies to the constant direction $v = x_1 - x_0$. Consequently, the primal training objective minimizes the regression error:

$$\mathcal{L}_{\text{primal}} = \mathbb{E}_{t, x_0, x_1} \left[ \|\hat{v}(x_t) - (x_1 - x_0)\|_2^2 \right]. \qquad (5)$$

Once trained, samples $\hat{x}_0$ can be recovered from the prior $x_1$ using numerical integration schemes.

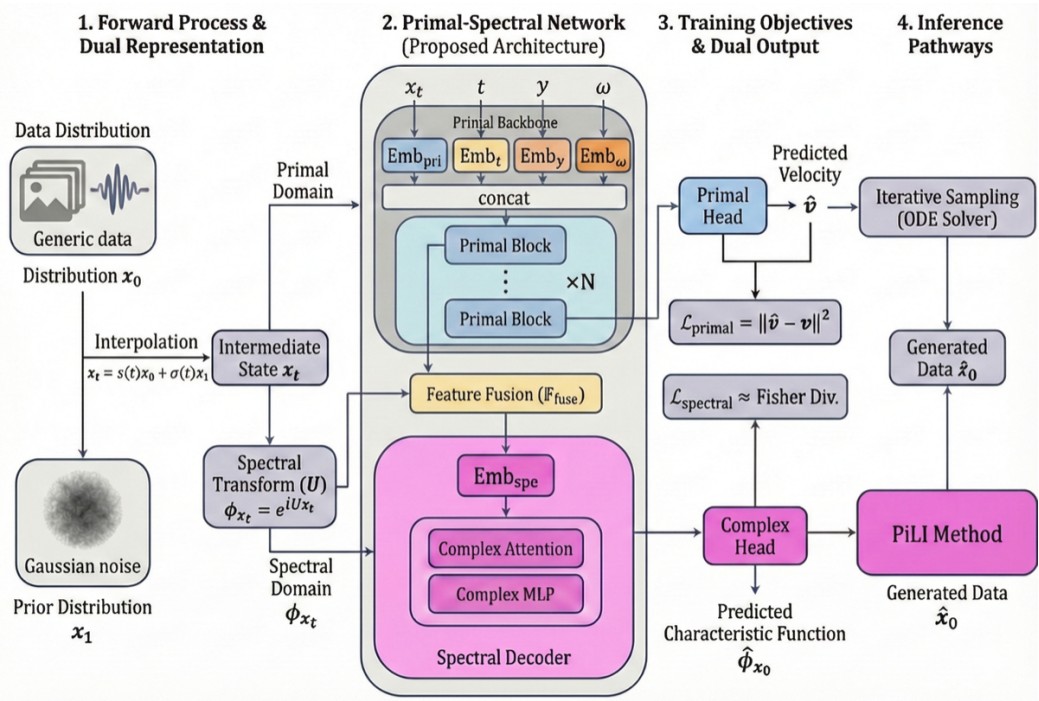

*Figure 2.* **The overview of Primal-Spectral Generative Model.** The system consists of four components: (1) A Dual Representation module that maps data into both primal state space and spectral domain; (2) A Primal-Spectral Network featuring a deep primal backbone fused with a complex-valued spectral decoder; (3) Training strategy optimizing $\mathcal{L}_{\text{primal}}$ and $\mathcal{L}_{\text{spectral}}$; and (4) Two distinct inference pathways: standard iterative ODE sampling and our proposed 1-NFE PiLI method for analytical generation.

## 4. Method

To overcome the limitations of primal-only methods, we propose the Primal-Spectral Generative Model, which is supported by three mathematically grounded pillars. Firstly, Forward Spectral Modulation transforms stochastic interpolants into tractable multiplicative phase modulations, capturing high-order correlations. Secondly, Reverse Spectral Reconstruction minimizes a spectral loss derived from two-sided Fisher divergence bounds, explicitly squeezing the distribution gap for accelerated convergence. Finally, PiLI is introduced to replace iterative solvers, a close-form formulation that enables high-fidelity, analytical 1-NFE generation.

### 4.1. Forward Spectral Modulation

Given a data vector $x_t \in \mathbb{R}^d$, the Spectral Feature Map projects the data into the spectral domain by evaluating the complex exponential basis at a set of $K$ vectors, organized as a matrix $U = [u_1, \ldots, u_K]^\top \in \mathbb{R}^{K \times d}$. Formally, this maps the input vector $x_t$ to a complex-valued spectral vector:

$$\phi_{x_t}(U) = [\phi_{x_t}(u_1), \ldots, \phi_{x_t}(u_K)]^\top \in \mathbb{C}^K, \qquad (6)$$

where $\phi_{x_t}(u_k) = e^{iu_k^\top x_t}$. Unlike the marginal case, this operation captures the spatial correlations within $x_t$ through the projection $u_k^\top x_t$. Following the stochastic interpolant coefficients defined in Eq. (1), the forward admits a multiplicative

representation in the spectral domain.

**Proposition 4.1.** *(Forward Spectral Modulation) Define the forward interpolant $x_t = s(t)x_0 + \sigma(t)x_1$, where $x_1$ is a latent variable drawn from an arbitrary prior distribution $P_1$. Given the matrix $U \in \mathbb{R}^{K \times d}$, the spectral representation satisfies the factorization property:*

$$\phi_{x_t}(U) = \phi_{s(t)x_0}(U) \odot \phi_{\sigma(t)x_1}(U), \qquad (7)$$

*or equivalently for each component $k \in \{1, \ldots, K\}$:*

$$\phi_{x_t}(u_k) = e^{is(t)u_k^\top x_0} \cdot e^{i\sigma(t)u_k^\top x_1}, \qquad (8)$$

*where $\odot$ denotes the element-wise product.*

This property confirms that the linearity in state space translates to a multiplicative phase modulation in the spectral domain, independent of the specific form of the prior $P_1$. The proof in Appendix B establishes that the translation in state space corresponds to a phase shift in spectral space.

### 4.2. Reverse Spectral Reconstruction

In this study, we squeeze the Fisher divergence of the probability distributions between lower and upper bounds. This enables alignment of the learned process with the true process by minimizing a spectral loss function, derived as a corollary of Proposition 4.2.

**Proposition 4.2.** *(Reverse Spectral Reconstruction) Let $p(x)$ and $\hat{p}(x)$ be two probability densities in $\mathbb{R}^d$ with corresponding CFs $\phi_x(u)$ and $\hat{\phi}_x(u)$. Assume the densities are bounded within the effective support: $0 < m \le p(x), \hat{p}(x) \le M < \infty$, and the data is bounded: $\sup_x \|\nabla_x p(x)\|_2 = C < \infty$. Define the spectral gradient difference $\mathscr{A}$ and spectral density difference $\mathscr{B}$:*

$$\mathscr{A} = \frac{1}{(2\pi)^d} \int_{\mathbb{R}^d} \|u\|_2^2 |\Delta\phi(u)|^2 \, du, \qquad (9)$$

$$\mathscr{B} = \frac{1}{(2\pi)^d} \int_{\mathbb{R}^d} |\Delta\phi(u)|^2 \, du, \qquad (10)$$

*where $\Delta\phi(u) = \hat{\phi}_x(u) - \phi_x(u)$. Then the Fisher divergence admits the following explicit two-sided bounds:*

$$\left( \frac{\sqrt{m}}{M} \sqrt{\mathscr{A}} - \frac{C}{m^{3/2}} \sqrt{\mathscr{B}} \right)^2 \le \text{Fisher}\left(\hat{p}(x) \| p(x)\right)$$

$$\le 2 \left( \frac{M}{m^2} \mathscr{A} + \frac{C^2}{m^3} \mathscr{B} \right). \qquad (11)$$

Minimizing the spectral terms $\mathscr{A}$ and $\mathscr{B}$ directly pushes down the upper bound of Fisher$(\hat{p}(x)\|p(x))$, guaranteeing convergence, thereby effectively mitigating the vanishing gradient and missing topological structure issues inherent illustrated in Fig. 1. Consequently, the spectral error $\Delta\phi(U)$ serves as a strong proxy for the intractable Fisher divergence. We explicitly minimize the spectral gradient difference $\mathscr{A}$ via a weighted objective. In practice, we approximate the continuous integral over spectral space using a grid matrix $U \in \mathbb{R}^{K \times d}$, leading to the discrete spectral loss:

$$\mathscr{L}_{\text{spectral}} = \mathbb{E}_{t,x_0,x_t} \left[ \frac{1}{K} \sum_{k=1}^{K} \|u_k\|_2^2 \left| \phi_{x_0}(u_k) - \hat{\phi}_{x_0}(u_k) \right|^2 \right] \qquad (12)$$

$$= \mathbb{E}_{t,x_0,x_t} \left[ \frac{1}{K} \left\| \Lambda \odot \left( \phi_{x_0}(U) - \hat{\phi}_{x_0}(U) \right) \right\|_2^2 \right], \qquad (13)$$

where $\Lambda = [\|u_1\|_2, \ldots, \|u_K\|_2]^\top$ is the weighting vector. By prioritizing the minimization of $\mathscr{A}$ through the $\|u_k\|_2^2$ terms, $\mathscr{L}_{\text{spectral}}$ effectively squeezes the Fisher divergence, aligning the learned spectral statistics with the true data distribution.

### 4.3. Pseudoinverse Lévy Inversion

We bypass iterative integration by formulating $x_0$ recovery as a weighted least squares problem with Tikhonov regularization $\lambda I$. This acts as a low-pass filter suppressing noise-dominated singular values, directly yielding a 1-NFE closed-form solution, which we term Pseudoinverse Lévy Inversion (PiLI):

$$\hat{x}_0 = \underbrace{(U^\top W U + \lambda I)^{-1} U^\top W}_{\text{Pseudoinverse}} \theta, \qquad (14)$$

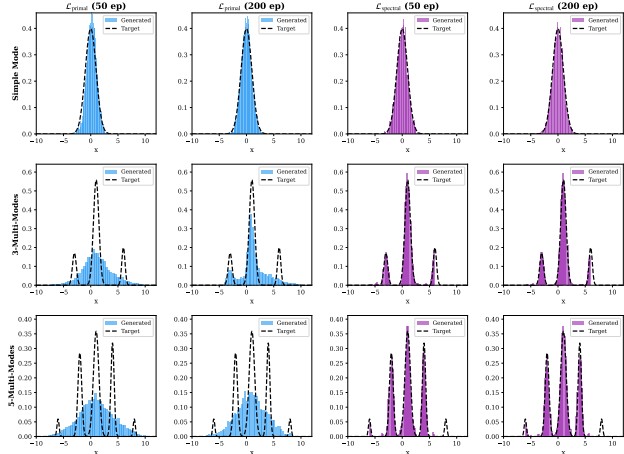

*Figure 3.* **Convergence across Gaussian Mixture Models.** 1) Simple Mode: $\mathcal{N}(0,1)$; 2) 3-Mode: means $\mu = [-3, 1, 6]$, scales $\sigma = [0.35, 0.5, 0.3]$, and weights $w = [0.15, 0.7, 0.15]$; and 3) 5-Mode: $\mu = [-6, -2, 1, 4, 8]$, $\sigma = [0.2, 0.35, 0.5, 0.3, 0.2]$, with highly skewed weights $w = [0.03, 0.25, 0.45, 0.24, 0.03]$. We use the same MLP architecture and parameter counts across 50 and 200 training epochs, the $\mathscr{L}_{\text{primal}}$ baseline with 50-NFE Heun's sampling exhibits significant over-smoothing and underfitting on complex multimodal distributions. In contrast, the $\mathscr{L}_{\text{spectral}}$ method with 1-NFE PiLI precisely resolves all modes.

where $W = \text{diag}(\Lambda \odot \Lambda)$ encodes the spectral consistency weights and $\theta = \arg(\phi_{\hat{x}_0}(U))$ represents the phase vector, and a detailed derivation is provided in Appendix C. To guarantee numerical stability and prevent phase wrapping, the frequency matrix $U$ is initialized as a fixed matrix with constant row norms $\|u_k\|_2 = \pi/\sqrt{d}$. Practically, with $x \in [-1, 1]$, this ensures $u_k^\top x \in [-\pi, \pi]$, circumventing the phase wrapping problem. Furthermore, this constant norm ensures that minimizing the gradient difference $\mathscr{A}_{\text{empirical}}$ strictly minimizes the density difference $\mathscr{B}_{\text{empirical}}$:

$$\mathscr{A}_{\text{empirical}} = \frac{1}{K} \sum_{k=1}^{K} \|u_k\|_2^2 |\Delta\phi(u_k)|^2 \propto \mathscr{B}_{\text{empirical}} \qquad (15)$$

We acknowledge the gap between the continuous integral over $\mathbb{R}^d$ and constant-norm sampling. However, fixing the norm is an acceptable engineering practice to avoid phase wrapping while allowing PiLI weighting to accommodate dynamic frequency importance. In practice, we efficiently solve this linear system using Cholesky decomposition to avoid explicit matrix inversion.

### 4.4. Network Architecture

To incorporate complex-valued spectral features into standard real-valued neural architectures, we first establish a dual representation using Euler decomposition. For a data vector $x_t \in \mathbb{R}^d$ and a matrix $U \in \mathbb{R}^{K \times d}$, the spectral representation $\phi_{x_t}(U)$ is explicitly split into real and imaginary

components:

$$\phi_{x_t}(U) = \underbrace{\cos(Ux_t)}_{\Re(\phi_{x_t}(U))} + i \cdot \underbrace{\sin(Ux_t)}_{\Im(\phi_{x_t}(U))}. \quad (16)$$

This decomposition maps complex phase and magnitude information onto a real-valued manifold, enabling the stacking of these components as input features. Building on this representation, we propose the Primal-Spectral Network in Fig. 2, which we refer to as PriSpecNet. Consistent with the architecture of iMF (Geng et al., 2025b), the Primal Backbone comprises $N$ stacked blocks using standard self-attention and SwiGLU Feed-Forward Networks. Following iMF's efficient control data integration, we adopt a streamlined input strategy where the scalar inputs for time $t$, label $y$, and guidance scale $\omega$ are first projected into token representations via their respective embedding layers ($\text{Emb}_t, \text{Emb}_y, \text{Emb}_\omega$). These conditioning tokens are then concatenated to the beginning of the image token sequence, serving as global context for the network.

To bridge the primal and spectral domains, we inject latent features $\mathbf{E}_{\text{pri}}$ from early primal blocks into the spectral path via element-wise addition: $\mathbf{E}_{\text{spe}} = \text{Emb}_{\text{spe}}(\phi_{x_t}) \oplus \mathbb{F}_{\text{fuse}}(\mathbf{E}_{\text{pri}})$, where $\mathbb{F}_{\text{fuse}}$ is a complex linear projection. The Spectral Decoder then processes this context-rich complex representation $\mathbf{E}_{\text{spe}}$ using Complex Attention and MLP adapted from (Wang et al., 2025a). Finally, the network yields the predicted $\hat{\phi}_{x_0}$ and $\hat{v}$. Following the CFG paradigm introduced in Improved Mean Flow (Geng et al., 2025b), during training, the guidance scale $\omega$ is explicitly sampled from a log-uniform interval $[1.0, 8.0]$ and fed into the network to directly learn $v_{\text{cfg}}(x_t, t, y, \omega) = (\omega - 1) \cdot v_{\text{uncond}}(x_t, t, \varnothing, 1.0) + v_{\text{cond}}(x_t, t, y, \omega)$. Concurrently, for the spectral guidance, the network predicts the conditional form $\hat{\phi}_{x_0}(U; x_t, t, y, \omega)$ to match the true data CF under the guidance scale $\omega$. For iterative inference, we fix $\omega = 4.0$ and use the directly predicted $v_{\text{cfg}}$. For 1-NFE inference, we predict the conditional phase $\hat{\theta} = \arg(\hat{\phi}_{x_0}(U; x_t, t, y, \omega = 4.0))$ and feed it directly into the PiLI analytical inversion. This approach achieves conditional generation while avoiding the double-evaluation cost of CFG during inference.

### 4.5. Analysis of Convergence and Efficiency

To illustrate the rapid convergence of our method, we conduct experiments across Gaussian Mixture Models. The results in Fig. 3 clearly shows that the convergence of spectral method is much faster than primal method, which is the direct consequence of our spectral generative modeling. In the primal domain, optimization on sparse samples often suffers from vanishing gradients in low-density regions, leading to local stagnation. In contrast, minimizing the spectral terms $\mathscr{A}$ and $\mathscr{B}$ provides persistent gradient signals across the entire domain, and our spectral upper bound

(Eq. (11)) explicitly enforces a global topology alignment. This ensures that the learned $\hat{p}_0$ is pulled towards the target $p_0$ from a global perspective early in training, effectively circumventing the primal-only local minima inherent.

Furthermore, we emphasize that PiLI offers high efficiency among analytical solvers, which arises from a fundamental paradigm shift: replacing *temporal integration* with *algebraic inversion*. While standard solvers must sequentially integrate the ODE to traverse the trajectory, PiLI achieves a 1-NFE closed-form solution via Eq. (14). Intuitively, the over-determined projection ($K > d$) acts as a robust information buffer: the Tikhonov regularization effectively filters high-frequency noise during the inversion process, allowing the solver to analytically recover the clean data manifold.

## 5. Experiment

Experiments are conducted on NVIDIA H200 GPUs using PyTorch. We evaluate across two distinct domains: Time Series and ImageNet. Notably, the regularity assumptions in Proposition 4.2 are well-justified in these practical settings: normalizing data to $[-1, 1]$ enforces a compact support, which, combined with the inherent Gaussian mollification, strictly guarantees the bounded densities $m \le p(x) \le M$ and finite scores $C < \infty$. In the time series generation experiment, our 50-NFE Heun's method establishes a new SOTA in both Context-FID and MSE/MAE, with the 1-NFE PiLI method explicitly outperforming previous SOTA baselines like ARMD and Diffusion-TS. In the ImageNet $256 \times 256$ generation study, we validate model scalability and spectral benefits. Our 1-NFE PiLI-XL achieves a FID of 1.66. Although this slightly trails the 25-NFE DPM-Solver++ FID of 1.62, it delivers an approximate **170× reduction** in Gflops, reducing the computational cost from 4,436 to just 26.

### 5.1. Time Series Generation

**Setting.** With fixed $y = \varnothing$ and $\omega = 1$, we evaluate time series generation performance on four benchmarks characterized by varying feature dimensions $d$, as detailed in Tab. 6: the low-dimensional Sines ($d = 5$), ETTh ($d = 7$) (Zhou et al., 2021), and Stock ($d = 6$) (Yoon et al., 2019), alongside the high-dimensional Solar ($d = 137$) (Lai et al., 2018). Accordingly, the spectral dimension $K$ is adapted (ranging from 32 to 256) to maintain the over-determined condition relative to each $d$. generation quality is assessed via Context Frechet Inception Distance (Context-FID) (Jeha et al., 2022), while MSE/MAE utilizes a 96-step lookback to predict the subsequent 96 steps. Results denote mean ± std. dev. over 5 independent runs. PriSpecNet-TS benchmark against three SOTA generative models: ARMD (Gao et al., 2025), a continuous sequential diffusion model replacing Gaussian noise with a deterministic sliding mechanism via a linear-based devolution network; TSFlow (Kollovieh et al.,

*Table 2.* **Generation and forecasting comparison.** Generation quality is assessed via Context-FID with a context length 24. MSE/MAE is evaluated for forecasting using a 96-length lookback window to predict the subsequent 96-length. Results denote mean $\pm$ std.

| Metric | Methods | Sines | Solar | ETTh | Stock |
|---|---|---|---|---|---|
| Context-FID ↓ | ARMD (Gao et al., 2025) | 0.006 ± .000 | 0.041 ± .003 | 0.093 ± .009 | 0.141 ± .013 |
| | TSFlow (Kollovieh et al., 2024) | 0.007 ± .001 | 0.077 ± .008 | 0.157 ± .021 | 0.201 ± .023 |
| | Diffusion-TS (Yuan & Qiao, 2024) | 0.006 ± .000 | 0.052 ± .006 | 0.116 ± .010 | 0.147 ± .015 |
| | PriSpecNet-TS (50-NFE Heun's) | **0.003 ± .000** | **0.024 ± .001** | **0.018 ± .000** | **0.052 ± .018** |
| | PriSpecNet-TS (1-NFE PiLI) | 0.006 ± .001 | 0.037 ± .003 | 0.033 ± .009 | 0.093 ± .021 |
| MSE ↓ | ARMD (Gao et al., 2025) | 0.016 ± .003 | 0.161 ± .027 | 0.430 ± .040 | 0.235 ± .023 |
| | TSFlow (Kollovieh et al., 2024) | 0.021 ± .003 | 0.183 ± .032 | 0.456 ± .041 | 0.338 ± .028 |
| | Diffusion-TS (Yuan & Qiao, 2024) | 0.027 ± .004 | 0.193 ± .038 | 0.468 ± .042 | 0.389 ± .031 |
| | PriSpecNet-TS (50-NFE Heun's) | **0.010 ± .001** | **0.113 ± .024** | **0.201 ± .034** | **0.179 ± .021** |
| | PriSpecNet-TS (1-NFE PiLI) | 0.016 ± .004 | 0.167 ± .054 | 0.250 ± .078 | 0.213 ± .044 |
| MAE ↓ | ARMD (Gao et al., 2025) | 0.017 ± .003 | 0.233 ± .028 | 0.465 ± .045 | 0.268 ± .029 |
| | TSFlow (Kollovieh et al., 2024) | 0.021 ± .003 | 0.243 ± .044 | 0.470 ± .047 | 0.359 ± .036 |
| | Diffusion-TS (Yuan & Qiao, 2024) | 0.028 ± .004 | 0.249 ± .045 | 0.471 ± .047 | 0.412 ± .042 |
| | PriSpecNet-TS (50-NFE Heun's) | **0.012 ± .002** | **0.177 ± .033** | **0.223 ± .036** | **0.189 ± .027** |
| | PriSpecNet-TS (1-NFE PiLI) | 0.015 ± .005 | 0.201 ± .041 | 0.256 ± .070 | 0.231 ± .061 |

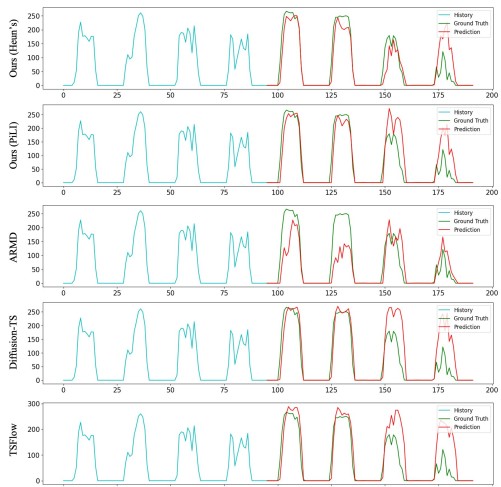

*Figure 4.* Visualizations of time series forecasting on Solar dataset

2024), a Conditional Flow Matching framework integrating DiffWave (Kong et al., 2020), S4 layers, and Gaussian Process priors via optimal transport; and Diffusion-TS (Yuan & Qiao, 2024), an interpretable Transformer-based model employing seasonal-trend decomposition and spectral-based loss to guide reconstruction. Further implementation details are provided in Appendix D.1.

**Results.** For the Context-FID task, we employ the 50-NFE Heun's method and the 1-NFE PiLI method. For prediction, we design a training-free framework, details shown on Algorithm 1 and Algorithm 2 in Appendix D.3, that treats forecasting as a constrained generation process, utilizing the iterative mode or PiLI mode. As reported in Tab. 2, PriSpecNet-TS (50-NFE Heun's) establishes a new SOTA across all datasets, specifically reducing Context-FID

*Table 3.* Network configurations for image generation.

| Config | Primal Blocks | Hidden Dim | Heads | #Params (M) |
|---|---|---|---|---|
| B | 16 | 768 | 12 | 136 |
| L | 24 | 1024 | 16 | 338 |
| XL | 32 | 1280 | 16 | 671 |

by 80.6% on ETTh and 63.1% on Stock compared to the strongest baseline ARMD, while also dominating long-term forecasting by lowering MSE on ETTh by 53.2% and on Stock by 23.8%. Remarkably, PriSpecNet-TS (1-NFE PiLI) secures the second-best performance globally and Fig. 4 visualizes the forecasting on Solar.

### 5.2. Image Generation

**Experiment Setting.** With $\omega$ sampled from a log-uniform distribution on $[1.0, 8.0]$ (Geng et al., 2025b) and class labels $y$ spanning the $1,000$ categories of ImageNet (Deng et al., 2009), we conduct our image generation experiments at a resolution of $256 \times 256$. Our models are implemented directly in pixel space and evaluated using FID computed on 50,000 generated samples. The Base (B), Large (L), and Extra-Large (XL) architectures introduced as PriSpecNet-B/L/XL in Fig. 2 process inputs with a patch size of 16. Crucially, we define the data dimension $d = 16 \times 16 \times 3$ as the flattened patch dimension. This configuration ensures that the spectral dimension $K$ satisfies the over-determined condition $K > d$ required for the stable inversion of the PiLI solver. All models in Tab. 3 are trained from scratch, regardless the primal and spectral models share the same number of hidden dimensions and attention heads. Further details are provided in Appendix E.1.

**1-NFE Generation Comparison.** We present a compre-

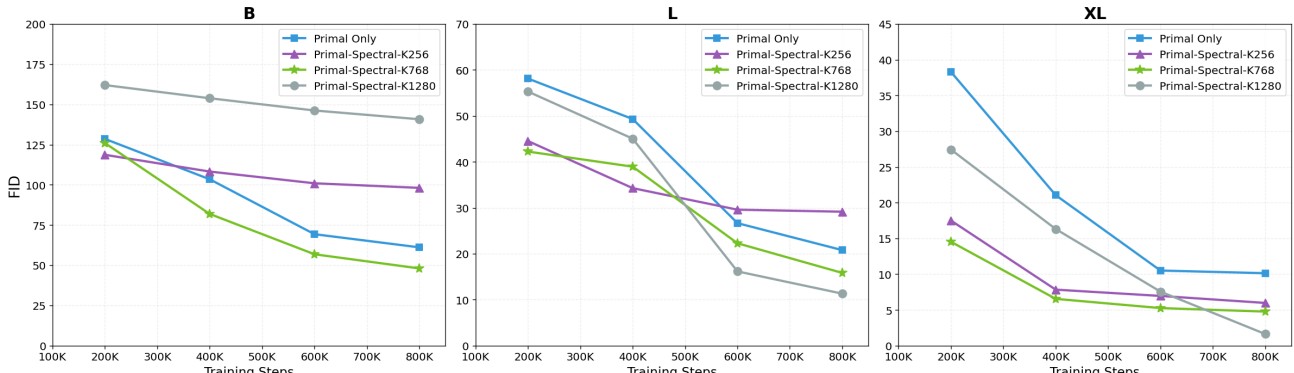

*Figure 5.* **Ablation study on model scaling and spectral dimensions.** We evaluate the training progression of PriSpecNet-B/L/XL models, reporting FID scores at intervals of 200k training steps. The curves compare performance across different $K \in \{256, 768, 1280\}$ for the matrix $U$. All evaluations are performed using Heun's method with 50 NFEs based on the predicted velocity $v_{cfg}$. The "Primal Only" baseline refers to the network architecture without the spectral components.

*Table 4.* **1-NFE Generation Comparison**. We report parameter counts and FID scores for models trained from scratch, distillation-based approaches, and GANs on ImageNet $256 \times 256$.

| Method | #Params (M) | NFE | FID↓ |
|---|---|---|---|
| *1-NFE generative model from scratch* | | | |
| iCT-XL/2 (Zhou et al., 2025) | 675 | 1 | 34.24 |
| Shortcut-XL/2 (Frans et al., 2024) | 675 | 1 | 10.60 |
| MeanFlow-XL/2 (Geng et al., 2025a) | 676 | 1 | 3.43 |
| TiM-XL/2 (Wang et al., 2025b) | 664 | 1 | 3.26 |
| $\alpha$-Flow-XL/2+ (Zhang et al., 2025a) | 676 | 1 | 2.58 |
| iMF-XL/2 (Geng et al., 2025b) | 610 | 1 | 1.72 |
| PiLI-XL (Ours) | 676 | 1 | **1.66** |
| *1-NFE generative mode (distillation)* | | | |
| $\pi$-Flow-XL/2 (Chen et al., 2025a) | 675 | 1 | 2.85 |
| DMF-XL/2+ (Lee et al., 2025) | 675 | 1 | 2.16 |
| FACM-XL/2 (Peng et al., 2025) | 675 | 1 | 1.76 |
| *GANs* | | | |
| BigGAN (Brock et al., 2018) | 112 | 1 | 6.95 |
| GigaGAN (Kang et al., 2023) | 569 | 1 | 3.45 |
| StyleGAN-XL (Sauer et al., 2022) | 166 | 1 | 2.30 |

*Table 5.* **Solver Efficiency Benchmark.** Measurements are conducted on a NVIDIA H200 GPU with a batch size of 1. Time is reported as the wall-clock latency averaged over 5 synchronized runs following 2 warmup steps to ensure stability. Total Gflops denotes theoretical floating-point operations.

| Solver | Steps | NFE | Total Gflops↓ | Time (s)↓ | FID↓ |
|---|---|---|---|---|---|
| Euler Method | 100 | 100 | 17,744 | 1.643 | 1.70 |
| Euler Method | 50 | 50 | 8,872 | 0.822 | 1.95 |
| Heun's Method | 50 | 100 | 17,566 | 1.627 | 1.70 |
| Heun's Method | 25 | 50 | 8,695 | 0.806 | 1.72 |
| DPM-Solver++ | 25 | 25 | 4,436 | 0.412 | **1.62** |
| RK45 | 35 | 140 | 24,842 | 2.303 | 1.66 |
| PiLI | **1** | **1** | **26** | **0.017** | 1.66 |

tive time-step integration to progressively generate samples. Conversely, our PiLI method circumvents this computationally expensive process by exploiting the efficient spectral path, which allows for direct, non-iterative data recovery. This unique mechanism significantly reduces the computational overhead, enabling PiLI to achieve generation with merely **26** Gflops—an approximate **170×** efficiency improvement compared to the 4,436 Gflops required by the 25-NFE DPM-Solver++.

**Model Scaling Ablation.** We conducted an extensive ablation study over 800k training steps across PriSpecNet-B/L/XL, targeting varying matrix dimensions $K \in \{256, 768, 1280\}$ and details are shown on Fig. 5. The results reveal the significant impact of dimension $K$ on model scaling. For the PriSpecNet-B model, $K = 256$ yields an early FID of 118.64 at 200k steps, slightly outperforming the 'Primal Only' baseline of 128.53 FID; however, constrained by limited expressivity, it stagnates in later stages, converging to 98.09 FID, which is significantly worse than the baseline. Conversely, $K = 1280$ creates a training bottleneck, preventing effective convergence with FID remaining above 140. Only $K = 768$ balances training efficiency with

hensive benchmarking of our PriSpecNet-XL model trained from scratch compared to SOTA 1-NFE generative frameworks on ImageNet $256 \times 256$ in Tab. 4. Remarkably, our method achieves a SOTA FID of **1.66**. This performance not only surpasses widely adopted GAN baselines and competitive distillation-based techniques, but also outperforms the previous best training-from-scratch baseline, iMF-XL/2 (1.72 FID). These results validate that our spectral inversion approach can generate high-fidelity samples directly from the prior without relying on iterative numerical integration or complex multi-stage distillation methods.

**Primal vs Spectral Solver.** Tab. 5 presents a comprehensive efficiency analysis of various generative algorithms using the pre-trained PriSpecNet-XL. Conventional solvers, including Euler, Heun, and DPM-Solver++, are constrained by the need to traverse the full primal path, requiring itera-

feature expressivity, achieving comprehensive superiority over the baseline with a final FID of 48.00. A similar trend is observed in the PriSpecNet-L model, where $K = 256$ restricts the learning upper bound. The final performance of 29.14, which lags behind the 'Primal Only' baseline, indicates that the spectral bottleneck limits the potential of larger models. For PriSpecNet-XL, smaller $K$ values $(256, 768)$ accelerate early convergence by simplifying the optimization landscape, rapidly dropping FID to 17.50 and 14.53. However, this comes at the cost of spectral resolution, leading to performance stagnation due to limited expressivity. Conversely the high-dimensional setting $K = 1280$ exhibits slower initial convergence, its larger spectral capacity prevents information loss. This allows for sustained improvement, ultimately reaching a SOTA FID of **1.62**, validating that sufficient spectral bandwidth is crucial for high-fidelity modeling.

## 6. Conclusion and Outlook

We have introduced Primal-Spectral Generative Modeling, a theoretically grounded framework that bridges the gap between intractable empirical distributions in the primal space and their continuous representations in the spectral domain. By rigorously bounding the Fisher divergence through spectral density minimization, it resolves the topological ambiguity often encountered in learning from sparse samples. A pivotal contribution of this study is the PiLI solver, which reformulates generation as a closed-form problem, replacing computationally expensive iterative ODE integration with a fast 1-NFE closed-form analytical solution.

Extensive evaluations show that our PriSpecNet establishes new SOTA performance in time series modeling — reducing Context-FID by up to 80.6% on standard datasets; as well as enabling high-fidelity 1-NFE image synthesis on ImageNet with an approximate $170\times$ reduction in Gflops compared to advanced diffusion solvers. Ultimately, this framework offers a scalable alternative with potential to achieve extreme inference efficiency while strictly preserving data fidelity in pixel space. Nevertheless, we acknowledge that current implementations lack dedicated tools to control the correlations among the spectral dimension $K$, the data dimension $d$, and the network hidden dimensions. Looking ahead to practical applications, to fully leverage the efficiency of the PiLI solver, future network architectures could be designed as Spectral-Only frameworks, thereby eliminating primal redundancy for ultimate performance.

## Acknowledgments

This work was supported in part by the National Key R&D Program of China under Grants 2021YFA1000300 and 2021YFA1000302. This work was also supported in part by the Fundamental and Interdisciplinary Disciplines Breakthrough Plan of the Ministry of Education of China under Grant JYB2025XDXM411, by the Shanghai Key Laboratory of PMMP, by the Science and Technology Commission of Shanghai Municipality under Grant No. 22DZ2229014, and by the Key Laboratory of Mathematics and Engineering Applications, Ministry of Education. The authors also gratefully acknowledge the computing support provided by the Shanghai Innovation Institute.

## Impact Statement

This work studies fast generative modeling through primal-spectral representations and analytical inversion, with potential benefits for efficient synthesis and forecasting in scientific computing, time series analysis, and other resource-constrained applications. At the same time, generative models can be misused to produce deceptive synthetic content, and their outputs may inherit biases or artifacts from the training data. Our experiments are limited to standard benchmarks and do not by themselves establish safety, fairness, or robustness in high-stakes deployments. We therefore recommend that practical use of such models be accompanied by careful dataset governance, transparent disclosure of synthetic content, and downstream safeguards appropriate to the target application.

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

## A. Spectral Form of Forward Interpolant

**Proposition 4.1.(Forward Spectral Modulation)** Define the forward interpolant $x_t = s(t)x_0 + \sigma(t)x_1$, where $x_1$ is a latent variable drawn from an arbitrary prior distribution $P_1$. Given the matrix $U \in \mathbb{R}^{K \times d}$, the spectral representation satisfies the factorization property:

$$\phi_{x_t}(U) = \phi_{s(t)x_0}(U) \odot \phi_{\sigma(t)x_1}(U), \tag{17}$$

or equivalently for each component $k$:

$$\phi_{x_t}(u_k) = e^{is(t)u_k^\top x_0} \cdot e^{i\sigma(t)u_k^\top x_1}, \tag{18}$$

where $\odot$ denotes the element-wise (Hadamard) product. This signifies that the linear mixing of data sample and noise in the data space corresponds to a rotation of the phase by a random noise phase in the spectral domain.

*Proof.* Let $U = [u_1, \ldots, u_K]^\top \in \mathbb{R}^{K \times d}$ denote the matrix, where each row $u_k \in \mathbb{R}^d$ is a vector. We analyze the $k$-th component of the spectral feature map, denoted as $\phi_{x_t}(u_k)$. By definition Eq. 6, the $k$-th component is given by:

$$\phi_{x_t}(u_k) = e^{iu_k^\top x_t}. \tag{19}$$

Substituting the definition of the forward stochastic interpolant, $x_t = s(t)x_0 + \sigma(t)x_1$, into the exponent, and exploiting the linearity of the inner product, we obtain:

$$\phi_{x_t}(u_k) = e^{iu_k^\top(s(t)x_0 + \sigma(t)x_1)} \tag{20}$$

$$= e^{is(t)u_k^\top x_0 + i\sigma(t)u_k^\top x_1} \tag{21}$$

$$= e^{is(t)u_k^\top x_0} \cdot e^{i\sigma(t)u_k^\top x_1}. \tag{22}$$

Since this factorization holds for every component $k = 1, \ldots, K$ independently, the relationship for the full spectral vector $\phi_{x_t}(U)$ is the element-wise (Hadamard) product of the spectral vector and the noise spectral vector:

$$\phi_{x_t}(U) = \phi_{s(t)x_0}(U) \odot \phi_{\sigma(t)x_1}(U). \tag{23}$$

This confirms that the additive Gaussian noise in the state space transforms into multiplicative phase noise in the spectral domain. $\square$

## B. Reversion in the Spectral Domain

### B.1. Fokker-Planck Equation in the Spectral Domain

While the data $x_t$ typically exists in a high-dimensional feature space $\mathbb{R}^d$, we model the evolution of its marginal distribution statistics via the CF evaluated at a set of $U = [u_1, \ldots, u_K]^\top \in \mathbb{R}^{K \times d}$. Starting from the general Fokker-Planck equation adapted to the process in Eq. (1), the evolution for a single spectral component corresponding to $u_k$ is governed by:

$$\frac{\partial \phi_{x_t}(u_k)}{\partial t} = \frac{\dot{s}(t)}{s(t)} \left( u_k^\top \nabla_{u_k} \phi_{x_t}(u_k) \right) - \left( \sigma(t)\dot{\sigma}(t) - \frac{\dot{s}(t)}{s(t)}\sigma(t)^2 \right) \|u_k\|^2 \phi_{x_t}(u_k). \tag{24}$$

Here, $\nabla_{u_k}$ denotes the gradient with respect to $u_k$, and $\|u_k\|^2$ is the squared Euclidean norm. Since this relationship holds for every row $u_k$ of the matrix $U$, we can aggregate these $K$ independent equations into a single vectorized system.

**Vectorization of the Drift Term.** The drift corresponds to a transport effect in the spectral domain. We represent the vectorized radial derivative operation directly as the column vector formed by the dot products for each component:

$$\text{Drift}_U = \left[ u_k^\top \nabla_{u_k} \phi_{x_t}(u_k) \right]_{k=1}^K \in \mathbb{C}^K. \tag{25}$$

This term represents the sensitivity of the phase along the direction of each vector.

**Vectorized Spectral Evolution Equation.** Combining these terms, the full dynamics of the CF vector are governed by the following system of ODE for $U$:

$$\frac{\partial \phi_{x_t}(U)}{\partial t} = \underbrace{\frac{\dot{s}(t)}{s(t)} \left[ u_k^\top \nabla_{u_k} \phi_{x_t}(u_k) \right]_{k=1}^K}_{\text{Spectral Transport}} - \underbrace{\left( \sigma(t)\dot{\sigma}(t) - \frac{\dot{s}(t)}{s(t)}\sigma(t)^2 \right) \left( [\|u_k\|^2]_{k=1}^K \odot \phi_{x_t}(U) \right)}_{\text{Spectral Damping}}. \tag{26}$$

This vector equation explicitly describes how the entire spectrum evolves: the spectrum undergoes transport driven by the radial derivatives, and functional damping determined by the squared frequency norms, where high-frequency components decay quadratically faster than low-frequency components.

## B.2. Squeeze Fisher Divergence

**Proposition 4.2. (Reverse Spectral Reconstruction)** Let $p(x)$ and $\hat{p}(x)$ be two probability densities on $\mathbb{R}^d$ with corresponding CFs $\phi_x(u)$ and $\hat{\phi}_x(u)$. Assume the densities are bounded within the effective support: $0 < m \le p(x), \hat{p}(x) \le M < \infty$, and the data score is bounded: $\sup_x \|\nabla_x p(x)\|_2 = C < \infty$. Define the spectral gradient difference $\mathscr{A}$ and spectral density difference $\mathscr{B}$:

$$\mathscr{A} = \frac{1}{(2\pi)^d} \int_{\mathbb{R}^d} \|u\|_2^2 |\Delta\phi(u)|^2 \, du, \quad \mathscr{B} = \frac{1}{(2\pi)^d} \int_{\mathbb{R}^d} |\Delta\phi(u)|^2 \, du, \tag{27}$$

where $\Delta\phi(u) = \hat{\phi}_x(u) - \phi_x(u)$. Then the Fisher divergence admits the following explicit two-sided bounds:

$$\left( \frac{\sqrt{m}}{M} \sqrt{\mathscr{A}} - \frac{C}{m^{3/2}} \sqrt{\mathscr{B}} \right)^2 \le \text{Fisher}\,(\hat{p}(x)\|p(x)) \le 2 \left( \frac{M}{m^2} \mathscr{A} + \frac{C^2}{m^3} \mathscr{B} \right). \tag{28}$$

*Remark:* In practice, the integrals $\mathscr{A}$ and $\mathscr{B}$ are estimated via Monte Carlo integration using $U \in \mathbb{R}^{K \times d}$ defined in Section 4.1.

*Proof.* For brevity, we simplify notation: $s = \nabla \log p$, $\hat{s} = \nabla \log \hat{p}$, $\Delta p = \hat{p} - p$. The Fisher divergence is defined as the expected squared Euclidean distance between the score functions:

$$\text{Fisher}(\hat{p}\|p) = \int_{\mathbb{R}^d} \|\hat{s}(x) - s(x)\|_2^2 \, p(x) \, dx. \tag{29}$$

First, we decompose the score difference algebraically. Since $\nabla\hat{p} = \nabla(p + \Delta p) = \nabla p + \nabla\Delta p$:

$$
\begin{aligned}
\hat{s} - s &= \frac{\nabla\hat{p}}{\hat{p}} - \frac{\nabla p}{p} = \frac{p\nabla\hat{p} - \hat{p}\nabla p}{p\hat{p}} \\
&= \frac{p(\nabla p + \nabla\Delta p) - (p + \Delta p)\nabla p}{p\hat{p}} \\
&= \frac{p\nabla\Delta p - \Delta p \nabla p}{p\hat{p}} = \underbrace{\frac{\nabla\Delta p}{\hat{p}}}_{A} - \underbrace{\frac{\Delta p}{\hat{p}} \frac{\nabla p}{p}}_{B}.
\end{aligned}
\tag{30}
$$
$$\tag{31}$$

Thus, the integrand is $\|A - B\|_2^2$. We now invoke Plancherel's theorem, which states that the $L^2$ norm of a function in the spatial domain is equal to the $L^2$ norm of its spectral transform. Applying this to the gradients:

$$\|\nabla\Delta p\|_2^2 = \int_{\mathbb{R}^d} \|\nabla_x \Delta p(x)\|_2^2 dx = \frac{1}{(2\pi)^d} \int_{\mathbb{R}^d} \|u\|_2^2 |\Delta\phi(u)|^2 du = \mathscr{A}. \tag{32}$$

Similarly for the density difference:

$$\|\Delta p\|_2^2 = \int_{\mathbb{R}^d} |\Delta p(x)|^2 dx = \frac{1}{(2\pi)^d} \int_{\mathbb{R}^d} |\Delta\phi(u)|^2 du = \mathscr{B}. \tag{33}$$

**Upper Bound.** Using the inequality $\|A - B\|_2^2 \le 2\|A\|_2^2 + 2\|B\|_2^2$, the divergence is bounded by:

$$\text{Fisher}(\hat{p}\|p) \le 2 \int \|A\|_2^2 p \, dx + 2 \int \|B\|_2^2 p \, dx. \tag{34}$$

For the first term, using the bounds on density:

$$\int \|A\|_2^2 p \, dx = \int \frac{\|\nabla\Delta p\|_2^2}{(\hat{p})^2} p \, dx \le \left( \sup_x \frac{p}{(\hat{p})^2} \right) \|\nabla\Delta p\|_2^2 \le \frac{M}{m^2} \mathscr{A}. \tag{35}$$

For the second term:

$$\int \|B\|_2^2 p \, dx = \int \frac{\|\nabla p\|_2^2}{p(\hat{p})^2} (\Delta p)^2 \, dx \le \left( \sup_x \frac{\|\nabla p\|_2^2}{p(\hat{p})^2} \right) \|\Delta p\|_2^2 \le \frac{C^2}{m^3} \mathscr{B}. \tag{36}$$

Combining these yields the upper bound.

**Lower Bound.** By the reverse triangle inequality in $L^2(p)$, we have $(\int \|A - B\|_2^2 p)^{1/2} \ge |(\int \|A\|_2^2 p)^{1/2} - (\int \|B\|_2^2 p)^{1/2}|$. We lower-bound the $A$-term and upper-bound the $B$-term:

$$\int \|A\|_2^2 p \, dx \ge \left( \inf_x \frac{p}{(\hat{p})^2} \right) \|\nabla \Delta p\|_2^2 \ge \frac{m}{M^2} \mathscr{A}. \tag{37}$$

Combining with the previously derived upper bound for the $B$-term we obtain:

$$\sqrt{\text{Fisher}} \ge \frac{\sqrt{m}}{M} \sqrt{\mathscr{A}} - \frac{C}{m^{3/2}} \sqrt{\mathscr{B}}. \tag{38}$$

Squaring both sides gives the explicit spectral lower bound. $\qquad \square$

## C. Derivation of Pseudoinverse Lévy Inversion

The spectral representation maps data $x_0 \in \mathbb{R}^d$ to the complex domain via $\phi_{x_0}(u_k) = e^{iu_k^\top x_0}$. The phase $\theta_k$ of this CF is linearly related to the data by the projection vector $u_k$:

$$\theta_k = \arg(\phi_{x_0}(u_k)) \equiv u_k^\top x_0. \tag{39}$$

In matrix form, let $\theta \in \mathbb{R}^K$ be the vector of predicted phases and $U \in \mathbb{R}^{K \times d}$ be the matrix. The relationship is $\theta \approx Ux_0$. To recover $x_0$ from the predicted spectral features, we formulate an optimization problem minimizing the weighted squared error between the projected phase $Ux$ and the target phase $\theta$. The weights $W$ prioritize spectral components based on the weighting vector $\Lambda$ defined in the spectral loss:

$$\min_x \mathscr{J}(x) = (Ux - \theta)^\top W (Ux - \theta), \tag{40}$$

where $W = \text{diag}(\Lambda \odot \Lambda)$ is a diagonal matrix containing the squared spectral weights. Note that for a fixed $U$ with constant row norms, this reduces to the unweighted objective. However, our weighted formulation accommodates dynamic or multi-scale $U$ where frequency importance varies. To ensure numerical stability and invertibility (especially when $U^\top W U$ is ill-conditioned), we introduce a **Tikhonov regularization** term $\lambda \|x\|_2^2$:

$$\mathscr{J}_{\text{reg}}(x) = (x^\top U^\top - \theta^\top) W (Ux - \theta) + \lambda x^\top x \tag{41}$$

$$= x^\top U^\top W U x - \theta^\top W U x - x^\top U^\top W \theta + \theta^\top W \theta + \lambda x^\top x \tag{42}$$

$$= x^\top (U^\top W U) x - 2(U^\top W \theta)^\top x + \text{const} + \lambda x^\top x. \tag{43}$$

Taking the gradient $\nabla_x \mathscr{J}_{\text{reg}}$ and setting it to zero to find the optimal $\hat{x}_0$:

$$\nabla_x \mathscr{J}_{\text{reg}} = 2U^\top W(Ux - \theta) + 2\lambda I x = 0 \tag{44}$$

$$\implies (U^\top W U + \lambda I)x = U^\top W \theta. \tag{45}$$

Solving for $x$ yields the analytical 1-NFE solution presented:

$$\hat{x}_0 = (U^\top W U + \lambda I)^{-1} U^\top W \theta. \tag{46}$$

## D. Experiments of Time Series Generative

### D.1. Implementation Details

Tab. 6 details the unified training configuration for time series generation across four datasets, where input-dependent parameters like feature size (5–137) and capacity $K$ (32–256) vary, but the core 1.78M-parameter architecture (6 layers, 8 heads) remains fixed. The PriSpecNet-TS is trained for 12,000 steps using the Adam optimizer with a Cosine Annealing schedule; this includes a *warmupLr* of $5.0 \times 10^{-5}$—the peak learning rate target reached after linearly increasing during the first 1,000 steps—before decaying to a minimum of $5.0 \times 10^{-6}$ to ensure stable convergence.

### D.2. Additional Experimental Results

*Table 6.* Implementation details of PriSpecNet-TS for time series generation.

| Configs | Sines | Solar | ETTh | Stock |
|---|---|---|---|---|
| Feature size | 5 | 137 | 7 | 6 |
| K | 32 | 256 | 64 | 32 |
| Total #Params (M) | | 1.78 | | |
| Primal Depth | | 6 | | |
| Hidden dim | | 128 | | |
| Heads | | 8 | | |
| Training Steps | | 12K | | |
| Batch Size | | 128 | | |
| Dropout | | 0.0 | | |
| Optimizer | | Adam | | |
| Lr Schedule | | Cosine Annealing With Warmup | | |
| Warmup | | 1K | | |
| Warmup Lr (Goyal et al., 2017) | | 5.0e-5 | | |
| Min Lr | | 5.0e-6 | | |
| EMA Decay | | 0.9999 | | |

To qualitatively and quantitatively evaluate the generation fidelity of the proposed model, we employ three distinct visualization techniques detailed in this section. First, the Kernel Density Estimation plots shown in Fig. 6 assess the alignment of marginal distributions. KDE is a non-parametric method that estimates the probability density function of the time series values. By smoothing the data histogram with a Gaussian kernel, it produces continuous curves where the x-axis represents the value magnitude and the y-axis represents density. The close overlap between the original (red) and synthetic (green) curves demonstrates that the model accurately captures the statistical range of the ground truth data. To quantify this distributional alignment, we compute the Kullback-Leibler (KL) divergence between the real and synthetic distributions. The consistently low KL divergence scores (averaging $< 0.05$ across datasets) quantitatively confirm that our model minimizes the distributional shift and faithfully reproduces the marginal statistics.

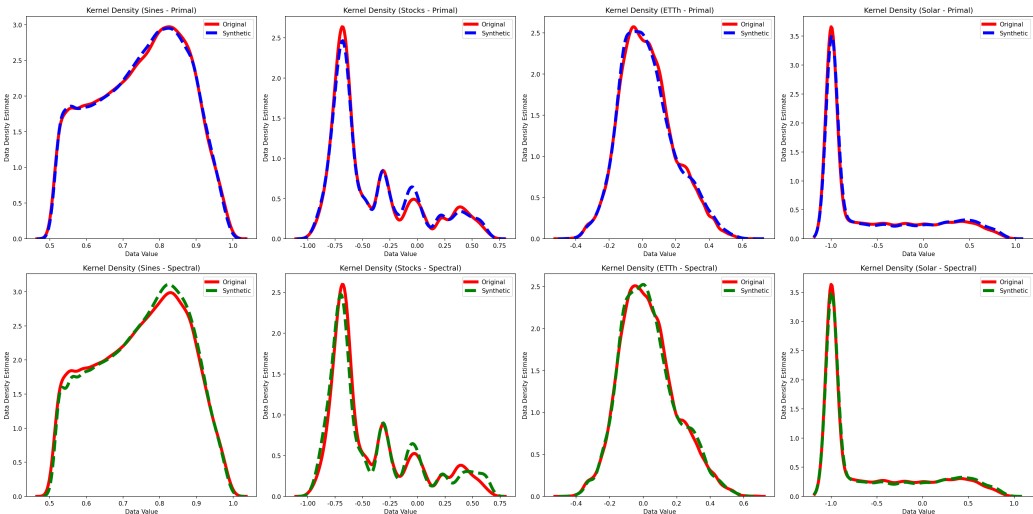

*Figure 6.* **Kernel Density Estimation** demonstrates that the synthetic marginal distributions (dashed) tightly align with the ground truth (solid). Notably, the model accurately reproduces the sharp skewness in the *Solar* dataset and the complex multi-modal densities in *Stocks* without exhibiting mode collapse.

Complementing the marginal analysis, Fig. 7 utilizes t-Distributed Stochastic Neighbor Embedding (t-SNE) to evaluate the preservation of the data's local geometric structure. As a non-linear dimensionality reduction technique, t-SNE maps

high-dimensional time series sequences onto a two-dimensional manifold while preserving local neighborhoods. The visualizations show a thorough intermixing of original and synthetic data points across all datasets. We rigorously validate this local structure preservation using the Adjusted Rand Index (ARI). The resulting low ARI scores indicate that the synthetic samples are indistinguishable from the real data within the local neighborhoods, successfully replicating the complex, non-linear clustering patterns and manifold topology of the real time series.

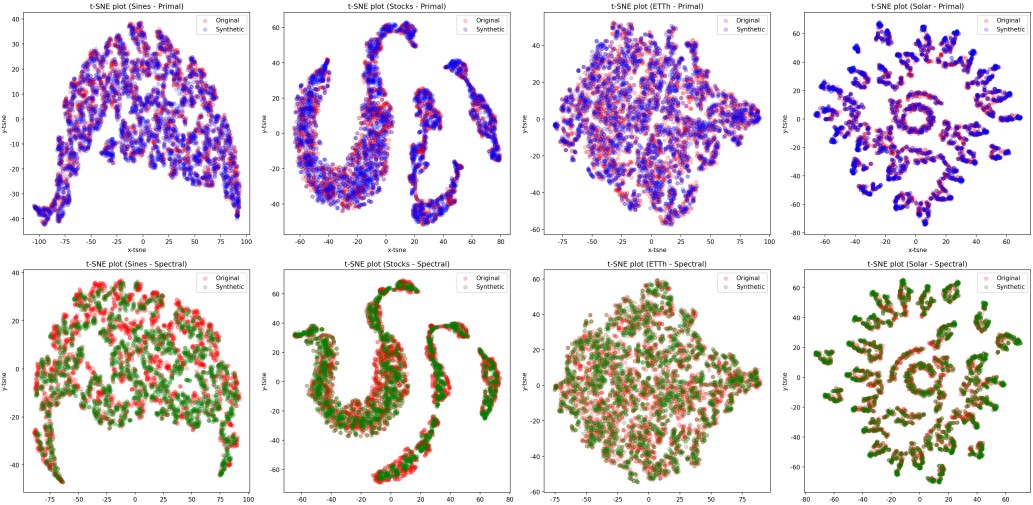

*Figure 7.* **t-SNE visualizations** reveal a thorough intermixing of original and generated points, confirming that our model preserves intricate non-linear manifolds and local neighborhood structures, successfully capturing the distinct clustering patterns observed in *ETTh* and *Stocks*.

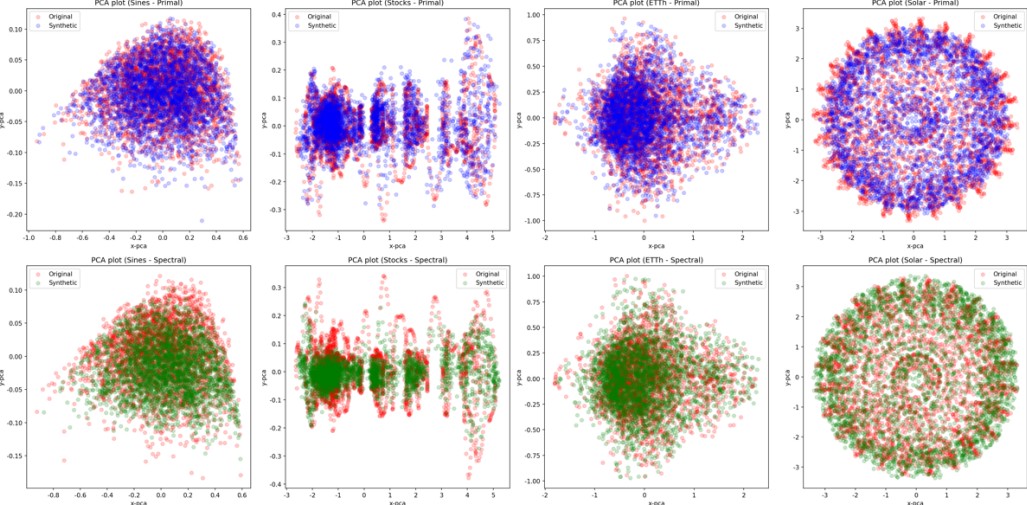

*Figure 8.* **PCA plots** show that the synthetic point clouds exhibit matching variance spreads and orientations along principal axes, validating the model's ability to retain global temporal correlations and linear dependencies across all benchmarks.

Finally, to verify the consistency of global temporal correlations, Fig. 8 presents visualizations based on Principal Component Analysis (PCA). Unlike t-SNE, PCA is a linear transformation that projects the high-dimensional data onto the axes of maximum variance (principal components). By plotting the data on the top two components, these scatter plots reveal the global variance structure. The matching shape, spread, and orientation of the synthetic and original point clouds are further corroborated by the Explained Variance Ratio (EVR). The negligible difference in cumulative EVR for the top principal components confirms that the model effectively retains the major global trends and linear dependencies inherent in the training data.

## D.3. Training-Free Time Series Forecasting

We propose the Primal-Spectral Conditional Generation Framework to recover data from partial observations $y_{\text{gt}}$ indicated by a binary mask $M$. Initialized from standard Gaussian noise $x_1 \sim \mathcal{N}(0, I)$, the framework operates in two distinct modes.

---

**Algorithm 1** Primal-Spectral PiLI Conditional Generation

---

1: **Input:** Observation $y_{\text{gt}}$, Mask $M$, Reg. $\lambda$
2: $x_1 \sim \mathcal{N}(0, I)$;  ▷ Initialize from standard Gaussian
3: $\_, \hat{\phi}_1 \leftarrow \text{Net}(x_1, 1)$;  ▷ Predict spectral features from prior
4: $\hat{x}_0 \leftarrow \text{Spectral\_Solve}(\hat{\phi}_1, \text{reg} = \lambda)$;  ▷ 1-NFE Analytical Inversion
5: $x_0 \leftarrow \text{Apply\_Constraint}(\hat{x}_0, y_{\text{gt}}, M)$;  ▷ Enforce observation consistency
6: **Return** $x_0$

---

In the latency-critical PiLI mode (Algorithm 1), the network first predicts spectral features $\hat{\phi}_1$ from the prior. Instead of iterative solving, we employ the Spectral\_Solve operator to analytically recover the clean data estimate $\hat{x}_0$ in a single step (1-NFE) via a regularized pseudoinverse formulation $(U^\top W U + \lambda I)^{-1}$. This is immediately followed by Apply\_Constraint, which enforces observation consistency through hard fusion: $M \odot y_{\text{gt}} + (1 - M) \odot \hat{x}_0$.

---

**Algorithm 2** Primal-Spectral Iterative Conditional Generation

---

1: **Input:** Obs. $y_{\text{gt}}$, Mask $M$, Steps $N$, Rate $\eta$, Weight $\lambda_s$
2: $x_1 \sim \mathcal{N}(0, I)$;  ▷ Initialize from standard Gaussian
3: **for** $i = N - 1$ **to** $0$ **do**
4:     $t \leftarrow i/N$;    $\hat{v}_t, \hat{\phi}_0 \leftarrow \text{Net}(x_t, t)$
5:     $\hat{x}_0 \leftarrow \text{Predict\_Clean\_State}(x_t, \hat{v}_t, t)$;  ▷ Estimate $x_0$ via flow
6:     **for** $k = 1$ **to** $K_{\text{guide}}$ **do**
7:         $\mathcal{L} \leftarrow \text{Calc\_Spectral\_Loss}(x_t, \hat{\phi}_0, y_{\text{gt}}, \lambda_s)$
8:         $x_t \leftarrow \text{Manifold\_Grad\_Step}(x_t, \mathcal{L}, \eta)$;  ▷ Guidance update
9:     **end for**
10:    $x_{\text{next}} \leftarrow \text{ODE\_Integrate}(x_t, \hat{v}_t, \Delta t = 1/N)$
11:    $x_{\text{traj}} \leftarrow \text{OT\_Interpolate}(x_1, y_{\text{gt}}, t)$
12:    $x_t \leftarrow \text{Apply\_Constraint}(x_{\text{next}}, x_{\text{traj}}, M)$;  ▷ Enforce boundary
13: **end for**
14: **Return** $x_t$

---

Conversely, the Iterative mode (Algorithm 2) performs numerical integration from $t = 1$ to $0$. Within each timestep, we first estimate the clean state via the flow. To refine this estimate, we execute a spectral guidance loop: calculating a composite loss $\mathcal{L}$ via Calc\_Spectral\_Loss—which balances spectral consistency against the predicted $\hat{\phi}_0$ and primal data fidelity—and updating the state $x_t$ using Manifold\_Grad\_Step. Finally, the state advances via ODE\_Integrate, while Apply\_Constraint ensures the observed components strictly follow the explicit optimal transport trajectory $x_{\text{traj}}$ (computed by OT\_Interpolate), guaranteeing that the final output $x_0$ is seamlessly coupled with the ground truth conditions.

# E. Experiments of Image Generation

## E.1. Implementation Details of Image

We instantiate our architecture across three model scales—Base (B), Large (L), and Extra-Large (XL)—to evaluate scalability. Detailed configurations are provided in Tab. 7. All models follow a consistent training protocol: we train for 800K steps with a global batch size of 256 using the Adam optimizer. A constant learning rate of $1 \times 10^{-4}$ (Goyal et al., 2017) is applied after a 10K-step linear warmup. To ensure stability and generation fidelity, we employ Exponential Moving Average (EMA) with a decay of 0.9999 and set the CFG (Ho & Salimans, 2022) conditioning dropout rate to 0.1. During training and inference, the CFG algorithm is implemented directly via the network's forward pass as detailed in Algorithm 3 and Algorithm 4.

---

**Algorithm 3** PriSpecNet: CFG Training

---

1: **Input:** Data $x_0$, Prior $x_1$, Label $y$, Matrix $U$
2: $t \sim \mathcal{U}(0,1)$
3: $x_t \leftarrow s(t)x_0 + \sigma(t)x_1$
4: $\omega \sim \text{Log-Uniform}(1.0, 8.0)$    ▷ Sample guidance scale
5: $\phi_0 \leftarrow \exp(iUx_0)$
6: $\phi_t \leftarrow \exp(iUx_t)$
7: $\hat{v}_{\text{pred}}, \hat{\phi}_{\text{pred}} \leftarrow \text{Net}(x_t, t, \phi_t, y, \omega)$
8: $\hat{v}_{\text{uncond}}, {}_{-} \leftarrow \text{Net}(x_t, t, \phi_t, \varnothing, \omega = 1.0)$
9: $\hat{v}_{\text{cond}} \leftarrow \hat{v}_{\text{pred}}.\text{detach}()$
10: $\hat{v}_{\text{cfg}} \leftarrow (\omega - 1.0) \cdot \hat{v}_{\text{uncond}} + \hat{v}_{\text{cond}}$
11: $\mathcal{L}_{\text{primal}} \leftarrow \text{MSE}(\hat{v}_{\text{pred}}, \hat{v}_{\text{cfg}})$    ▷ Implicit CFG target
12: $\mathcal{L}_{\text{spectral}} \leftarrow \text{Weighted\_MSE}(\hat{\phi}_{\text{pred}}, \phi_0)$    ▷ Match CF under scale $\omega$
13: $\mathcal{L} \leftarrow \mathcal{L}_{\text{primal}} + \mathcal{L}_{\text{spectral}}$
14: **Return** $\mathcal{L}$

---

**Algorithm 4** PriSpecNet: Iterative ODE and 1-NFE PiLI Inference

---

1: **Input:** Prior $x_1$, Label $y$, Matrix $U$, Target $\omega = 4.0$

2: *# Iterative ODE Inference*
3: **def** model_fn($x_t, t$):
4:     $\phi_t \leftarrow \exp(iUx_t)$
5:     $\hat{v}_{\text{pred}}, {}_{-} \leftarrow \text{Net}(x_t, t, \phi_t, y, \omega)$    ▷ Direct velocity prediction
6:     **return** $\hat{v}_{\text{pred}}$
7: $\hat{x}_0^{\text{ODE}} \leftarrow \text{ODE\_Solver}(\text{model\_fn}, x_{\text{init}} = x_1, \text{span} = [1.0, 0.0])$

8: *# 1-NFE PiLI Inference*
9: $\phi_1 \leftarrow \exp(iUx_1)$
10: ${}_{-}, \hat{\phi}_{\text{pred}} \leftarrow \text{Net}(x_1, t = 1.0, \phi_1, y, \omega)$    ▷ Direct spectral prediction
11: $\hat{\theta} \leftarrow \arg(\hat{\phi}_{\text{pred}})$
12: $\hat{x}_0^{\text{PiLI}} \leftarrow (U^\top W U + \lambda I)^{-1} U^\top W \hat{\theta}$    ▷ Analytical Inversion
13: **Return** $\hat{x}_0^{\text{ODE}}, \hat{x}_0^{\text{PiLI}}$

---

### E.2. Additional Experimental Results

We visualize the class-conditional generation results of our 1-NFE PiLI solver on ImageNet-256 across diverse categories, including Animals (Fig. 9), Objects (Fig. 10), Plants (Fig. 11), Scenes (Fig. 12), and Vehicles (Fig. 13).

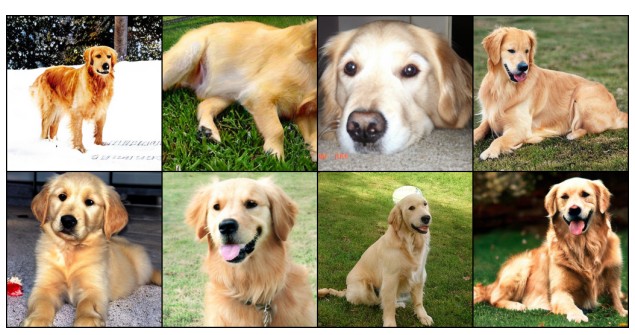 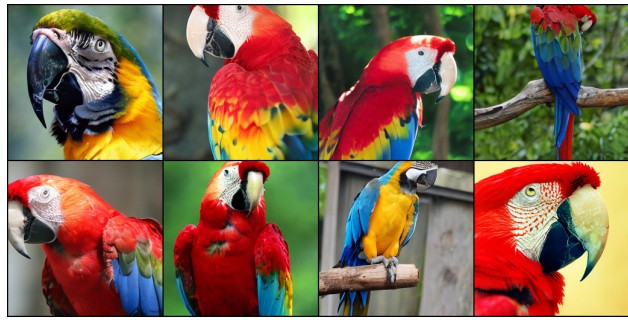

*(a)* class 207: golden retriever                *(b)* class 88: macaw

*Figure 9.* **Qualitative results on Animals.** Generated samples for Golden Retriever and Macaw using PiLI on ImageNet-256.

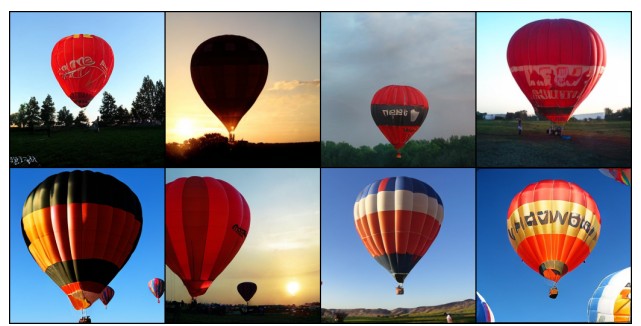

*(a)* class 417: balloon

*(b)* class 928: ice cream

*Figure 10.* **Qualitative results on Objects.** Generated samples for Balloon and Ice Cream using PiLI on ImageNet-256.

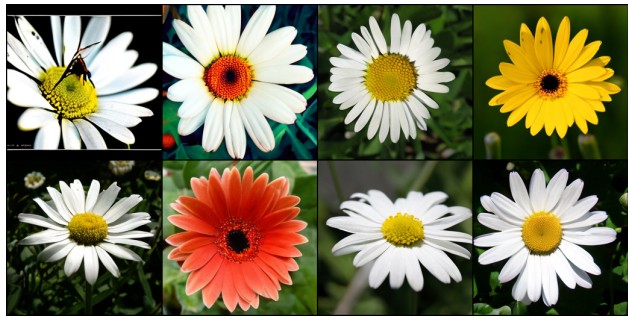
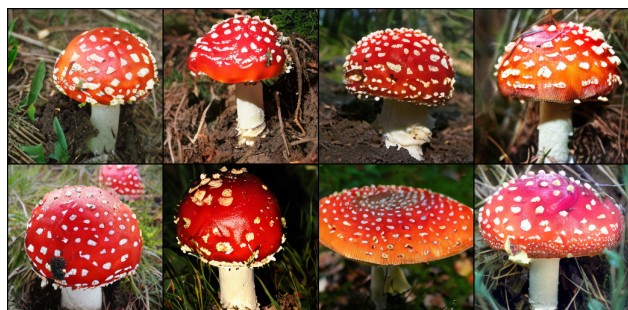

*(a)* class 985: daisy

*(b)* class 992: agaric mushroom

*Figure 11.* **Qualitative results on Plants.** Generated samples for Daisy and Agaric Mushroom using PiLI on ImageNet-256.

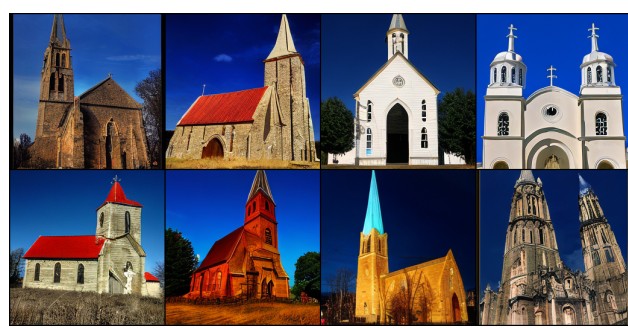
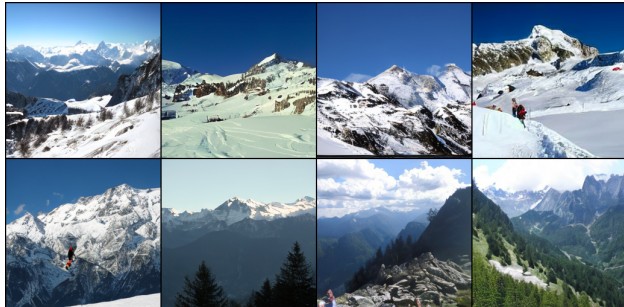

*(a)* class 497: church

*(b)* class 970: alp

*Figure 12.* **Qualitative results on Scenes.** Generated samples for Church and Alp using PiLI on ImageNet-256.

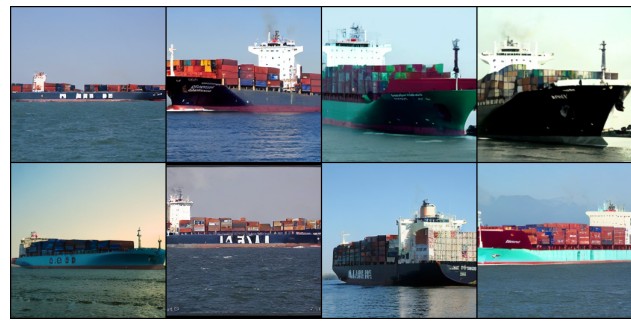
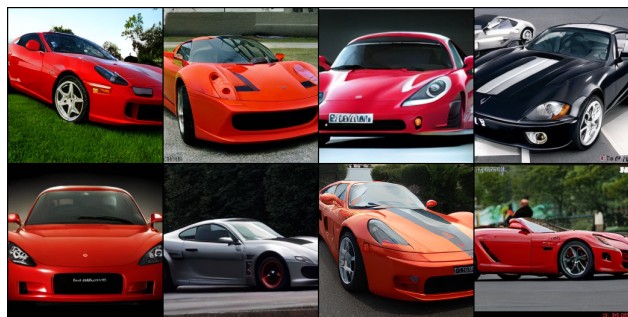

*(a)* class 510: container ship

*(b)* class 817: sports car

*Figure 13.* **Qualitative results on Vehicles.** Generated samples for Container Ship and Sports Car using PiLI on ImageNet-256.

*Table 7.* Implementation details of PriSpecNet-X/L/XL for image generation.

| Configs | B | L | XL |
|---------|-----|------|------|
| Total #Params (M) | 136 | 338 | 671 |
| Primal Depth | 16 | 24 | 32 |
| Hidden dim | 768 | 1024 | 1280 |
| Heads | 12 | 16 | 16 |
| Training Steps | | 800K | |
| Batch Size | | 256 | |
| Dropout | | 0.0 | |
| Optimizer | | Adam | |
| Lr Schedule | | Constant | |
| Warmup | | 10K | |
| Learning Rate (Goyal et al., 2017) | | 1e-4 | |
| EMA Decay | | 0.9999 | |
| CFG Cond Drop (Ho & Salimans, 2022) | | 0.1 | |

### E.3. K-Nearest Neighbors Analysis to Rule Out Memorization

To conclusively demonstrate that PriSpecNet does not merely memorize the training dataset, we conduct a *k*-Nearest Neighbors (k-NN) analysis for our generated samples. We evaluate the similarity between our generated samples and the original ImageNet training set in the deep feature space using a pre-trained ResNet-50 backbone, which captures semantic structures better than pixel-level distance. Specifically, for randomly selected generated samples across various classes, we retrieve the top-5 closest images from the respective class in the training set based on cosine similarity.

*Table 8.* **Quantitative Memorization Check Results.** We report the cosine similarities between generated samples and their top-5 nearest neighbors in the ImageNet training set using ResNet-50 features.

| Class Index | Top-1 Sim | Top-5 Mean Sim | Top-5 Max Sim | Top-5 Min Sim |
|-------------|-----------|----------------|---------------|---------------|
| 88 | 0.9401 | 0.9386 | 0.9401 | 0.9376 |
| 207 | 0.9384 | 0.9204 | 0.9384 | 0.9095 |
| 279 | 0.9086 | 0.8305 | 0.9086 | 0.7733 |
| 360 | 0.8816 | 0.8712 | 0.8816 | 0.8671 |
| 387 | 0.9694 | 0.9661 | 0.9694 | 0.9619 |
| 417 | 0.9614 | 0.9533 | 0.9614 | 0.9461 |
| 974 | 0.9412 | 0.9352 | 0.9412 | 0.9316 |
| 979 | 0.9101 | 0.9036 | 0.9101 | 0.8969 |

As shown in Tab. 8, while the top nearest training images share similar high-level semantics, the maximum cosine similarities remain largely below 0.97, and the mean similarities of the top-5 neighbors are even lower. Furthermore, visual inspections of the nearest neighbors confirm distinct structural differences in terms of object pose, background composition, and fine-grained textures. This substantial variance rigorously confirms that our 1-NFE PiLI solver genuinely learns and samples from the underlying continuous probability manifold mapped in the spectral domain, synthesizing novel distributions rather than suffering from data regurgitation (memorization).

