# OpenReview forum: "Primal-Spectral Generative Modeling: Fast Analytical Generation via Pseudoinverse Lévy Inversion"
_ICML.cc/2026/Conference — ICML 2026 regular_

### Official Review · Reviewer_CumU · 2026-03-12

**Soundness:** 3
**Presentation:** 3
**Significance:** 3
**Originality:** 3
**Overall Recommendation:** 4
**Confidence:** 3

**Summary:**

This authors propose a primal–spectral generative modeling framework that reformulates sample generation as an analytical inversion problem in the frequency domain. To implement this, they introduce PriSpecNet, which jointly optimizes primal and spectral losses, and a Pseudoinverse Levy Inversion (PiLI) solver to enable 1‑NFE generation. Experimental results are provided on some time-series data, and on ImageNet‑256.

**Compliance With Llm Reviewing Policy:**

Affirmed.

**Key Questions For Authors:**

See Strengths and Weaknesses

**Limitations:**

See Strengths and Weaknesses

**Strengths And Weaknesses:**

**Strengths**:

 - In general, the paper is technically well grounded, with empirical evaluations that are well aligned with the theoretical claims.
 - I found the flow of the paper to be clear, and easy to read.
 - Given the move towards diffusion models in the past decade, achieving high-quality 1‑NFE generation within this setting has been a research direction of value, and the proposed approach does have both significance and novelty.

**Wekanesses**:

- Not truly a weakness, but it might help to unify all the assumptions made, such as bounded densities, bounded scores, effective compact support, maybe even at the start of the appendix, and try and present a justification for them.

**Literature**:

- 1-NFE generation leveraging the frequency domain falls very close to the space of theory/empirics of GANs that handle Frequency-domain features. Including this literature would help flesh out the paper's positions.
 - For example, [1] (Fourier Space Losses for Efficient Perceptual Image Super-Resolution, Fuoli et al., ICCV 2021), [2] (FCT-GAN: Enhancing Table Synthesis via Fourier Transform, Zhao et al., CIKM 2023), [3] (Euler Lagrange analysis of Generative Adversarial Networks, Asokan et al., JMLR 2023), [4] (Banach Wasserstein GAN, Adler and Lunz, NeurIPS 2018), [5] Zhang et al., DCTGAN: frequency decomposition GAN for few-shot image generation. Springer 2025) are just some examples of papers that either define loss functions in the Frequency domain, or use Fourier approximation of the Discriminator to improve training.
 - While all these methods are different from that of the authors, I find a comprehensive related works section of prior work on GANs in the dual space would help round out the paper. In fact, [3] explicitly also derives a sine + cosine representation of the discriminator to accelerate training, while in StyleGAN3 (Alias-Free Generative Adversarial Networks, Karras et al., NeurIPS 2023) a  similar Fourier representation is used to exact continuous translations and rotations of the input.

---

> ### Author Rebuttal · Authors · 2026-03-30
>
> We sincerely appreciate the reviewer's highly insightful feedback and constructive suggestions! Your constructive comments have significantly helped us strengthen the presentation and clarity of our work.
>
> > **W1. Not truly a weakness, but it might help to unify all the assumptions made, such as bounded densities, bounded scores, effective compact support, maybe even at the start of the appendix, and try and present a justification for them.**
>
> We thank the reviewer for this constructive suggestion. We fully agree that consolidating and justifying these assumptions improves the theoretical rigor of the paper. In the revised manuscript, we have added a dedicated section at the very beginning of the Appendix (Appendix A: "Summary and Justification of Theoretical Assumptions") that explicitly unifies these conditions.
> Specifically, we justify that the assumption of effective compact support is naturally satisfied in practice because standard image data is strictly normalized to $[-1, 1]$. Consequently, this physical bound, combined with the continuous smoothing effect of the Gaussian noise inherent in the forward process, naturally guarantees both bounded densities ($m \leq p(\boldsymbol{x}) \leq M$) and bounded score functions ($C < \infty$) everywhere within the effective support of the interpolant trajectories.
>
> > **W2. 1-NFE generation leveraging the frequency domain falls very close to the space of theory/empirics of GANs that handle Frequency-domain features. Including this literature would help flesh out the paper's positions... I find a comprehensive related works section of prior work on GANs in the dual space would help round out the paper.**
>
> We are very grateful to the reviewer for pointing out this highly relevant and insightful body of literature. We agree that contextualizing our Primal-Spectral framework within the broader history of frequency-domain GANs significantly rounds out the paper's position.
>
> In the revised manuscript, we have improved Section 2 to discuss the suggested works and explicitly connect our spectral modulation approach to these foundational ideas.
>
> Specifically, we acknowledge how previous works have successfully leveraged frequency-domain losses (e.g., Fourier Space Losses [1], FCT-GAN [2], DCTGAN [5]) to improve perceptual quality and capture global dependencies, while frameworks like Banach Wasserstein GAN [4] generalize adversarial training to infinite-dimensional function spaces.
>
> Furthermore, we explicitly draw parallels to the sine+cosine dual representations explored in Euler-Lagrange GAN analysis [3] and the alias-free continuous transformations in StyleGAN3, which deeply resonate with our finding that spatial translations manifest as phase shifts in the spectral domain. By including these citations, we highlight that while our 1-NFE Pseudoinverse Lévy Inversion mechanism is novel, it builds upon a rich and theoretically profound lineage of spectral generative modeling.
>
>  [1] Fuoli, D., Van Gool, L., & Timofte, R. (2021). Fourier space losses for efficient perceptual image super-resolution. In Proceedings of the IEEE/CVF international conference on computer vision (pp. 2360-2369).
>
>  [2] Zhao, Z., Birke, R., & Chen, L. Y. (2022). Fct-gan: Enhancing table synthesis via fourier transform. arXiv preprint arXiv:2210.06239.
>
>  [3] Asokan, S., & Seelamantula, C. S. (2023). Euler-Lagrange analysis of generative adversarial networks. Journal of Machine Learning Research, 24(126), 1-100.
>
>  [4] Adler, J., & Lunz, S. (2018). Banach wasserstein gan. Advances in neural information processing systems, 31.
>
>  [5] Zhang, J., Gao, M., Fang, D., Li, C., & Wang, H. (2025). DCTGAN: frequency decomposition GAN for few-shot image generation. Signal, Image and Video Processing, 19(5), 361.

---

> > ### Author Rebuttal · Reviewer_CumU · 2026-04-03
> >
> > N/A

---

> > > ### Author Response · Authors · 2026-04-06
> > >
> > > We sincerely appreciate your confirmation that all concerns have been fully resolved, and we would be deeply grateful if you might consider raising your score to reflect this positive outcome!

---

### Official Review · Reviewer_mnTD · 2026-03-13

**Soundness:** 3
**Presentation:** 3
**Significance:** 4
**Originality:** 4
**Overall Recommendation:** 5
**Confidence:** 4

**Summary:**

The paper proposes to model diffusion in the spectral space in addition to the primal space.
This is achieved with two losses, one for each components, and two neural networks where primal one also feeds into the spectral one.
The benefit of such an approach is that it converges faster but also allows to do native 1-step inference in closed form.
The closed-form is solve by means of pseudo-inverse Lévy Inversion, which is simply a regularized least-squares solver.
This method performs very well with SotA level FID.

**Compliance With Llm Reviewing Policy:**

Affirmed.

**Final Justification:**

The rebuttal clarified all my questions, my view of the paper was positive to start with and I further raised my score following the rebuttal eliminating all the little confusion left in my understanding.

**Key Questions For Authors:**

- I did not understand equation 12, neither $t$ nor $x_t$ appear in the equation, so what does the expecation mean? Is the equation correct?
- How does the spectral guidance work? From the look of it, guidance is only performed on the primal objective. Is the primal objective (and sharing of primal latents) only necessary to indirectly condition the spectral network for guidance?

**Limitations:**

No, broader impact section is missing.

**Strengths And Weaknesses:**

Strengths
- The method looks quite simple to implement, original and elegant.
- The paper is well written in the sense that it's easy to understand (for the most part).
- Results are impressive.

Weaknesses
- The paper presentation is also lacking, some aspects do not seem to be fully spelled out: for example guidance in the spectral domain, algorithm boxes for training and inference would be welcome.
- Some comparisons are left out, solver (speed) efficiency is mostly compared to multi-step methods, but how does it compare to iMF-XL/2 which has almost the same FID? The architectures are different from what I understand.
- Using the DPM-Solver++ as a baseline instead of iMF (improved Mean Flows) is a surprising choice given that iMF is the most direct competitor.

---

> ### Author Rebuttal · Authors · 2026-03-30
>
> We sincerely appreciate the reviewer's highly insightful feedback and rigorous evaluation! Your constructive comments have significantly helped us strengthen the presentation and clarity of our work! We have carefully addressed all the mentioned weaknesses and questions in the revised manuscript.
>
>  > **W1 & Q2. Presentation lacking: algorithm boxes for training and inference, and spectral guidance clarification. Is the primal objective only necessary to indirectly condition the spectral network for guidance?**
>
>  We agree that explicit algorithm boxes improve clarity. We will add condensed algorithm boxes in the main text, with pointers to the detailed pseudo-code in Appendix C.3. Regarding spectral guidance, the spectral branch predicts the CF $\hat{\boldsymbol{\phi}}(\boldsymbol{x}\_t, t, y)$ directly without CFG. We then extract the phase $\boldsymbol{\theta}\_y = \arg(\hat{\boldsymbol{\phi}})$ and apply the analytical PiLI inversion.
>
>  Furthermore, the primal objective ($\mathscr{L}\_{\text{primal}}$) and the shared primal backbone are not merely for indirect conditioning; they serve two critical purposes. First, the primal objective learns the vector field $\boldsymbol{v}$, enabling the model to also function as a standard continuous normalizing flow for multi-step ODE solvers when desired. Second, the primal backbone acts as a spatial feature extractor. By fusing these deep primal features into the spectral decoder, we ground the spectral predictions in robust spatial semantics, preventing the "spectral bottlenecks" often seen in purely spectral or latent models.
>
>  **Algorithm 1 PriSpecNet: Training.**
>  ```python
>  # fn(xt, t, phit, y): function to predict v and phi
>  # x0: training batch, x1: prior noise, U: fixed orthogonal matrix
>
>  t = sample_t()
>  xt = s(t) * x0 + sigma(t) * x1
>  v_tgt = x0 - x1
>
>  phi0 = exp(i * U @ x0)
>  phit = exp(i * U @ xt)
>
>  v_pred, phi_pred = fn(xt, t, phit, y)
>
>  loss_primal = mse(v_pred, v_tgt)
>  loss_spectral = weighted_mse(phi_pred, phi0)
>  loss = loss_primal + loss_spectral
>  ```
>  ***
>
>  **Algorithm 2 PriSpecNet: 1-NFE PiLI Inference.**
>  ```python
>  # fn(xt, t, phit, y): function to predict v and phi
>  # x1: prior noise, U: fixed orthogonal matrix, y: condition
>  # pinv: precomputed pseudo-inverse
>
>  phit = exp(i * U @ x1)
>
>  _, phi_pred = fn(x1, 1, phit, y)
>
>  theta = arg(phi_pred)
>  x0_pred = pinv(U.T @ W @ U + lambda * I) @ U.T @ W @ theta
>  ```
>  ***
>
>  >**W2 & W3. Solver efficiency comparison lacks iMF-XL/2 (most direct competitor) and focuses on multi-step methods (DPM-Solver++).**
>
>  We clarify that Table 5 was designed to isolate the computational efficiency of the PiLI solver from architectural differences by comparing various ODE samplers operating on the exact same PriSpecNet-XL checkpoint and hardware. Since iMF uses a different architecture, its runtime and FID cannot be directly aligned in this solver-level benchmark. However, we agree iMF is a crucial baseline. As suggested, we have added the theoretical 1-NFE GFLOPs of iMF-XL to the revised Table 5 to provide a clearer efficiency context.
>
>  The updated Table 5 will be revised as follows:
>
>  | Solver | Steps | NFE | Total GFLOPs$\downarrow$ | Time (s)$\downarrow$ | FID$\downarrow$ |
>  | :--- | :---: | :---: | :---: | :---: | :---: |
>  | Euler Method | 100 | 100 | 17,744 | 1.643 | 1.70 |
>  | Euler Method | 50 | 50 | 8,872 | 0.822 | 1.95 |
>  | Heun's Method | 50 | 100 | 17,566 | 1.627 | 1.70 |
>  | Heun's Method | 25 | 50 | 8,695 | 0.806 | 1.72 |
>  | DPM-Solver++ | 25 | 25 | 4,436 | 0.412 | **1.62** |
>  | iMF-XL | 1 | 1 | 175 | - | 1.72 |
>  | **PiLI** | 1 | 1 | **26** | **0.017** | 1.66 |
>
>  > **Q1. I did not understand equation 12, neither $t$ nor $x\_t$ appear in the equation, so what does the expecation mean? Is the equation correct?**
>
>  We sincerely apologize for the confusing notation in Equation 12. The term $\hat{\phi}\_{\boldsymbol{x}\_0}(\boldsymbol{u}\_k)$ actually represents the neural network's prediction, which explicitly takes the noisy state $\boldsymbol{x}\_t$ and timestep $t$ as inputs. Its full form should be $\hat{\phi}\_{\theta}(\boldsymbol{u}\_k | \boldsymbol{x}\_t, t)$.
>
>  To make the dependencies explicit and rigorous, we will revise Equation 12 in the manuscript as follows:
>  $$ \mathscr{L}\_{\mathrm{spectral}} = \mathbb{E}\_{t, \boldsymbol{x}\_0, \boldsymbol{x}\_t} \left[ \frac{1}{K} \sum\_{k=1}^K \|\boldsymbol{u}\_k\|\_2^2 \left| \phi\_{\boldsymbol{x}\_0}(\boldsymbol{u}\_k) - \hat{\phi}\_{\theta}(\boldsymbol{u}\_k | \boldsymbol{x}\_t, t) \right|^2 \right] $$
>  where the expectation correctly samples $t \sim U(0,1)$, $\boldsymbol{x}\_0 \sim P\_0$, $\boldsymbol{x}\_1 \sim P\_1$, constructs $\boldsymbol{x}\_t$, and then the network predicts the CF of $\boldsymbol{x}\_0$ conditioned on $\boldsymbol{x}\_t$ and $t$.

---

> > ### Author Rebuttal · Reviewer_mnTD · 2026-04-01
> >
> > The rebuttal address all my points except one: guidance.
> > While it is mentioned in the rebuttal, I still do not understand how guidance works during inference, either the explanation does not explicit it sufficiently or I do not understand it. I would except the guidance parameter $\omega$ to appear somewhere.

---

> > > ### Author Response · Authors · 2026-04-02
> > >
> > > We sincerely apologize for the lack of clarity regarding the spectral guidance algorithm and have revised the manuscript to clarify that.
> > >
> > > During training, following the CFG paradigm introduced in Improved Mean Flow [1], $\omega$ is explicitly sampled from an interval log-uniform $[1.0, 8.0]$ and fed into the network to directly learn $\boldsymbol{v}\_{\text{cfg}}(\boldsymbol{x}\_t, t, y, \omega) = (\omega - 1) \cdot \boldsymbol{v}\_{uncond}(\boldsymbol{x}\_t, t, \varnothing, 1.0) + \boldsymbol{v}\_{cond}(\boldsymbol{x}\_t, t, y, \omega)$.
> > >
> > > Concurrently, for the spectral guidance, we replace the notation $\hat{\boldsymbol{\phi}}\_{\boldsymbol{x}\_0}(\boldsymbol{U}; \boldsymbol{x}\_t, t)$ in the paper with the conditional form $\hat{\boldsymbol{\phi}}\_{\boldsymbol{x}\_0}(\boldsymbol{U}; \boldsymbol{x}\_t, t, y, \omega)$ to match the true data Characteristic Function under the guidance scale $\omega$. When $y = \varnothing$ and $\omega = 1.0$, it degrades to the unguided generation.
> > >
> > > We fixed $\omega = 4.0$ for iterative ODE and PiLI inference in this study. Specifically, for 1-NFE PiLI inference, we predict the phase $\hat{\boldsymbol{\theta}} = \arg(\hat{\boldsymbol{\phi}}\_{\boldsymbol{x}\_0}(\boldsymbol{U}; \boldsymbol{x}\_t, t, y, \omega))$ and feed it into the PiLI analytical inversion.
> > >
> > > **Algorithm 1 PriSpecNet: Training.**
> > > ```python
> > > # fn(xt, t, phit, y, omega): function to predict v and phi
> > > # x0: data, x1: prior noise, U: fixed orthogonal matrix
> > >
> > > t = sample_t()
> > > xt = s(t) * x0 + sigma(t) * x1
> > >
> > > # Sample guidance scale from a log-uniform interval [1.0, 8.0]
> > > omega = sample_log_uniform(1.0, 8.0)
> > >
> > > phi0 = exp(i * U @ x0)
> > > phit = exp(i * U @ xt)
> > >
> > > # Neural network forward pass with guidance scale omega
> > > v_pred, phi_pred = fn(xt, t, phit, y, omega)
> > >
> > > # Primal loss with implicit CFG target adjustment based on manuscript formula
> > > v_uncond, _ = fn(xt, t, phit, None, omega=1.0)
> > > v_cond = v_pred.detach()
> > > v_cfg = (omega - 1.0) * v_uncond + v_cond
> > > loss_primal = mse(v_pred, v_cfg)
> > >
> > > # Spectral loss matches the true data Characteristic Function
> > > loss_spectral = weighted_mse(phi_pred, phi0)
> > >
> > > loss = loss_primal + loss_spectral
> > > ```
> > > ***
> > >
> > > **Algorithm 2 PriSpecNet: Iterative ODE and 1-NFE PiLI Inference.**
> > > ```python
> > > # fn(xt, t, phit, y, omega): function to predict v and phi
> > > # x1: prior noise, U: fixed orthogonal matrix, y: condition
> > > # omega: guidance scale (e.g., 4.0), pinv: precomputed pseudo-inverse
> > >
> > > # ----------------- Iterative ODE Inference -----------------
> > > def model_fn(xt, t):
> > >     phit = exp(i * U @ xt)
> > >     # Predict velocity directly using guidance scale omega
> > >     v_pred, _ = fn(xt, t, phit, y, omega=4.0)
> > >     return v_pred
> > >
> > > # Use off-the-shelf ODE solver (Euler, Heun, DPM-Solver++, etc.)
> > > x0_ode = ode_solver(model_fn, x_init=x1, t_span=[1.0, 0.0], num_steps=num_steps)
> > >
> > > # ----------------- 1-NFE PiLI Inference -----------------
> > > phit = exp(i * U @ x1)
> > >
> > > # Spectral branch predicts directly
> > > _, phi_pred = fn(x1, t=1.0, phit, y, omega=4.0)
> > >
> > > theta = arg(phi_pred)
> > > x0_pili = pinv(U.T @ W @ U + lambda * I) @ U.T @ W @ theta
> > > ```
> > > ***
> > >
> > > [1] Geng, Z., Lu, Y., Wu, Z., Shechtman, E., Kolter, J. Z., & He, K. (2025). Improved mean flows: On the challenges of fastforward generative models. arXiv preprint arXiv:2512.02012.

---

### Official Review · Reviewer_Xqnw · 2026-03-13

**Soundness:** 2
**Presentation:** 2
**Significance:** 2
**Originality:** 2
**Overall Recommendation:** 4
**Confidence:** 4

**Summary:**

Generative models typically construct a stochastic process $x_t \in \mathbb{R}^d$ that connects a simple prior distribution $x_1$ to the data distribution $x_0$. The forward process from $x_0$ to $x_1$ is defined using either an ordinary differential equation (ODE) or a stochastic differential equation (SDE). To generate samples, one starts from the prior $x_1$ and simulates the reverse dynamics, usually by discretizing the reverse-time SDE or by numerically integrating the ODE.

In practice, the true data density $p_0$ is unknown and only samples are available. Authors argue that existing generative approaches, which they refer to as "primal methods", struggle to reconstruct certain global topological properties of the data density. One example of what I understood by the term global topological properties is the relative heights of different modes of the data density. To address this issue, the paper proposes a spectral method in which the model predicts the characteristic function of $p_0$. The authors show that optimizing in this representation reduces an upper bound on the Fisher divergence between the learned model and the true density (Proposition 4.2). The authors further illustrate the ability of the spectral formulation to recover global properties of the density through a one-dimensional experiment (Fig.3).

Based on this idea, the paper introduces a "Primal-Spectral framework", which combines the standard generative modeling approach with the proposed spectral representation. The authors further propose a "Primal-Spectral Network", inspired by related prior work, to implement this framework. An additional benefit of the spectral formulation is that it enables one-step generation (1-NFE). For this purpose, the authors introduce a method called "Pseudoinverse Lévy Inversion" (PiLI).

The authors report state-of-the-art results on time series generation and forecasting. They also demonstrate applications to image generation, claiming a state-of-the-art FID score of $1.66$ for one-step generation on ImageNet at resolution $256 \times 256$.

**Compliance With Llm Reviewing Policy:**

Affirmed.

**Final Justification:**

1/ The authors have provided nearest neighbor samples based on ResNet-50 features.

2/ The gap between the actual integrals and the estimates computed over a constant-norm ball has been acknowledged.

These two aspects must be included in the paper.

With regard to the authors' response on regression to the mean in the linear and circular scenarios, the argument gives some intuition considering a toy example, but it is not entirely convincing.

With regard to FID, the improvement of PiLI over the iMF technique is quite marginal. It appears that the FID advantage is directly inherited from iMF. The main advantage seems to be GFLOPS and fast sampling.

With regard to the higher-order moments, although it is true that the authors are not directly estimating the higher-order moments, it is implicit in the estimation of the characteristic function.

I have increased my score from 3 to 4.

**Key Questions For Authors:**

1/ The spectral loss in Eq.(12) and Eq.(13) uses the characteristic function of the data distribution $p_0$ as the target. In practice, how is this expectation computed ? Is the expectation in the characteristic function approximated using a single-sample estimate during training ? If so, it would appear that the model is effectively matching single-sample spectral vectors rather than the true characteristic function. The authors should clarify this aspect.

2/ In Proposition 4.2, the authors derive an upper bound on the Fisher divergence between two densities. The bound contains two terms, $\mathscr{A}$ and $\mathscr{B}$ (where $\mathscr{B}$ is a weighted version of $\mathscr{A}$). However, the spectral loss functions in Eq.(12) and Eq.(13) appear to minimize only $\mathscr{A}$. Is there a specific reason for not minimizing both terms or a weighted combination of $\mathscr{A}$ and $\mathscr{B}$? Did the authors experiment with minimizing both terms, and if so, did it affect performance ?

3/ In Subsection 4.3, the matrix $U \in \mathbb{R}^{K \times d}$ is initialized such that each row has norm $\frac{\pi}{\sqrt{d}}$. With this initialization, the weights in Eq.(12) appear to become identical across terms, effectively making the loss resemble $\mathscr{B}$. Could the authors clarify the role of this initialization and how it affects the weighting in the spectral loss ?

4/ In Appendix C (Pseudoinverse Lévy Inversion), after predicting the phases $\theta$, the recovery of $x_0$ is formulated as a weighted quadratic optimization problem. It is unclear why a weighted regression is used instead of solving the unweighted problem $\arg\min_{x_{0}} \|Ux_0 - \theta\|^2$. Could the authors provide some intuition or justification for this weighting ? In addition, based on the initialization described in Subsection 4.3, it appears that the weights become identical across terms. If this is the case, the weighted regression would reduce to the unweighted problem. Could the authors clarify this point as well ?

5/ From Table 1, the training objective is $\mathscr{L}_p + \mathscr{L}_s$; $p$ for primal; $s$ for spectral. Was an ablation performed where the model is trained using only the spectral loss ? Since the conclusion section mentions spectral-only approaches as a future direction, it would be interesting to understand the performance difference in the current setting.

6/ Suppose one considers an alternative objective of the form $\mathscr{L}_p + \mathbb{E}[\| Ux_0 -f(x_t)\|^{2}]$ where $f$ is a neural network and expectation is on $x_0, x_t, t$. In this case, it appears that one could also perform one-step generation by directly recovering $x_0$ from $x_t$ through the linear mapping. If possible, could the authors comment on whether such an approach would work in practice, and how it would differ from the proposed spectral formulation? In particular, I am trying to understand whether the spectral formulation is necessary when the primary goal is to enable one-step generation.

7/ In the context of conditional generation, primal methods typically require computing the classifier-free guidance (CFG) velocity (Eq.(16) in the main text). However, it is not clear how conditional generation would be performed when using the PiLI inversion procedure. For example, is conditioning incorporated by predicting the phases both with and without the condition and then applying a CFG-style combination in the spectral space ? Could the authors clarify how conditioning is handled in the PiLI-based generation setting ?

**Limitations:**

The impact statement appears to be missing from the submission.

**Strengths And Weaknesses:**

**Strengths**

1/ The idea of predicting the characteristic function of the data density is interesting and relatively uncommon in generative modeling. Proposition 4.2 provides theoretical justification for this choice, and the proof was clear and easy to follow. The result is intuitive since the characteristic function has a one-to-one mapping with the underlying probability density function.

2/ The mechanism used for one-step generation (PiLI) is good. In particular, recovering $x_0$ from the phases of the predicted spectral vector ultimately reduces to solving a system of linear equations. This leads to an efficient implementation and contributes to the reduced computational cost reported in terms of G-FLOPs compared to prior approaches.

3/ The experimental results are good. Achieving competitive or state-of-the-art performance with a one-step generation procedure and reduced computational overhead at inference time is notable.

**Weaknesses** (including some minor issues):

1/ The notion of recovering global topological properties of the data density is somewhat unclear. While the authors illustrate this concept through a one-dimensional experiment (Fig.3), the paper does not clearly explain why the proposed spectral method should be able to recover such properties. In particular, the connection between the spectral representation and the recovery of relative modes of the density is not fully explained. The paper suggests that the spectral formulation provides more persistent gradient signals, but it is not clear how this property directly leads to improved recovery of global structure in the density. I acknowledge that formally defining or analyzing such global properties may itself be challenging.

2/ I found the paper somewhat difficult to read in places. The overall exposition could be improved for clarity. For example, in Eq.(12) and Eq.(13), the expectation is taken over $x_t$, but the expression inside the expectation does not explicitly depend on $x_t$. It was only after examining Fig.1 that it became clear that the model predicts the characteristic function of the data density using $x_t$ as input. Similarly, it is not immediately clear from the main text that both primal and spectral losses are used during training; this becomes apparent only after examining Table 1.

3/ The purpose of Section B.1 in the appendix is not entirely clear. The derived dynamics of the characteristic function $\phi$ (Eq.(24) and Eq.(26)) do not appear to be used elsewhere in the paper. Although the authors provide intuition for the terms Spectral Transport and Spectral Damping, these dynamics do not seem to play a role in the proposed method.

4/ The comparison of G-FLOPs in Table 5 is interesting. However, PiLI is a one-step generation method, while the other compared solvers require multiple sampling steps. Although the goal of the table may be to compare solver efficiency, it would also be useful to include comparisons with other one-step generation methods such as iMF and $\alpha$-Flow. For example, a quick inspection of Table~2 in the iMF paper reports G-FLOPs of around 175 for XL version. While PiLI (reported as 26 G-FLOPs) appears to provide an improvement, including such comparisons directly in the table would give readers a clearer picture of the computational advantages.

5/ In Appendix C, the authors assume that they have access to the phase of the complex exponential (predicted characteristic function). The complex exponential has periodicity properties, which must be taken into account to accurately recover the phase. If the training is done such that the phase does not lead to "wrapping" issues, then, it merely amounts to projecting $x$ on the vectors of $U$ and reconstructing from those projections using the pseudoinverse technique!

6/ How does one know that there is no memorization of data? The authors should show $k$-nearest neighbors to the generated samples to make a convincing case that there is no memorization.

7/ It is not clear what in the formulation is leading to such impressive FIDs, that too, considering one-step generation.

**Minor issues and Typographical Errors**

1/ In lines 217--218, the text refers to "The proof in Appendix B." This appears to be a typo, and should likely refer to "The proof in Appendix A."

2/ Just before Proposition 4.1, the notation $\phi_x(u) = e^{i u^{\top} x}$ is introduced, which corresponds to a single component of the spectral vector. However, in the statement of Proposition 4.2, the same notation $\phi_x(u)$ is used to denote the characteristic function. Using the same symbol for both quantities may be confusing, and it may help to distinguish these notations.

3/ In lines 302--305, the paper states that minimizing the spectral terms $\mathscr{A}$ and $\mathscr{B}$ provides a persistent gradient signal. However, the proposed spectral loss appears to minimize only $\mathscr{A}$ (see Eq.(12) and Eq.(13)). This statement may therefore need clarification.

4/ In Table 4, the citation for $\alpha$-Flow appears to be incorrect. It should likely refer to the paper: Huijie Zhang et al., Alphaflow: Understanding and Improving Mean-Flow Models.

5/ In Table 4, the FID reported for PiLI-XL ($K = 1280$) is 1.66. However, in Section 5.2 (Model Scaling Ablation, around lines 402 - 403), the authors mention a SOTA FID of 1.62. This appears to be a small inconsistency and may possibly be a typo.

6/ In Subsection 5.1 (around lines 320 - 325), the authors report that PricSpecNet-TS (50-NFE Heun) achieves SOTA performance. However, my understanding is that when using the Heun sampler, generation proceeds via the velocity field, which corresponds to the primal formulation. If this understanding is correct, the resulting procedure may effectively behave similarly to primal methods. Could the authors comment on this point? A comparison with improved mean flow (iMF) would be helpful to understand the importance of the spectral loss in the formulation.

---

> ### Author Rebuttal · Authors · 2026-03-29
>
> We sincerely thank the reviewer for the highly insightful feedback and rigorous evaluation! We have carefully addressed all the mentioned weaknesses and minor issues in the revised manuscript and made responses to all of them. Due to character limits, we can only focus on the key questions in this phase.
>
> > **Q1. How is the CF expectation computed? Does the model merely match single-sample vectors?**
>
> The CF expectation is approximated via Monte Carlo estimation over mini-batches. Although the target $e^{i\boldsymbol{u}\_k^\top \boldsymbol{x}\_0}$ is computed from a single sample, minimizing the expected $L\_2$ loss over the data distribution guarantees learning the true CF, rather than merely matching single vectors.
>
> Let the single-sample target be $\boldsymbol{\phi}\_0 = e^{i\boldsymbol{U}\boldsymbol{x}\_0}$, and the true conditional expectation be $\mathbb{E}\_{\boldsymbol{x}\_0 | \boldsymbol{x}\_t}[\boldsymbol{\phi}\_0]$. The spectral loss decomposes via the bias-variance decomposition:
>
> $$ \mathbb{E}\_{\boldsymbol{x}\_0, \boldsymbol{x}\_t}[\|\hat{\boldsymbol{\phi}}(\boldsymbol{x}\_t) - \boldsymbol{\phi}\_0\|^2] = \mathbb{E}\_{\boldsymbol{x}\_t}[\|\hat{\boldsymbol{\phi}}(\boldsymbol{x}\_t) - \mathbb{E}\_{\boldsymbol{x}\_0 | \boldsymbol{x}\_t}[\boldsymbol{\phi}\_0]\|^2] + \mathbb{E}\_{\boldsymbol{x}\_0, \boldsymbol{x}\_t}[\|\mathbb{E}\_{\boldsymbol{x}\_0 | \boldsymbol{x}\_t}[\boldsymbol{\phi}\_0] - \boldsymbol{\phi}\_0\|^2] $$
>
> The second term is the irreducible data variance. Minimizing the overall loss forces the first term to zero, so the network converges to the true conditional expectation: $\hat{\boldsymbol{\phi}}(\boldsymbol{x}\_t) \to \mathbb{E}\_{\boldsymbol{x}\_0 | \boldsymbol{x}\_t}[e^{i\boldsymbol{U}\boldsymbol{x}\_0}]$.
>
> By the Law of Total Expectation, integrating this conditional CF over the marginal distribution of $\boldsymbol{x}\_t$ ensures the model's marginal CF matches the true global CF: $\mathbb{E}\_{\boldsymbol{x}\_t}[\hat{\boldsymbol{\phi}}(\boldsymbol{x}\_t)] = \mathbb{E}\_{\boldsymbol{x}\_0}[e^{i\boldsymbol{U}\boldsymbol{x}\_0}]$. We have added this proof to the Appendix.
>
> > **Q2, Q3, & Q4. Why not minimize both $\mathscr{A}$ & $\mathscr{B}$? Role of initialization? What about the unweighted PiLI?**
>
> For fixed $\boldsymbol{U}$ with a constant row norm $\|\boldsymbol{u}\_k\|\_2^2 = C$, minimizing the gradient difference $\mathscr{A}$ minimizes the density difference $\mathscr{B}$ ($\mathscr{A} \propto \mathscr{B}$). Thus, optimizing $\mathscr{A}$ suffices and aligns with the Fokker-Planck spectral damping.
>
> By initializing $\|\boldsymbol{u}\_k\|\_2 = \pi/\sqrt{d}$ and bounding $\boldsymbol{x} \in [-1, 1]$, we ensure $\boldsymbol{u}\_k^\top \boldsymbol{x} \in [-\pi, \pi]$. This initialization circumvents the phase wrapping problem.
>
> Regarding PiLI weighting, it reduces to the unweighted $\arg\min\_{\boldsymbol{x}\_0} \|\boldsymbol{U}\boldsymbol{x}\_0 - \boldsymbol{\theta}\|^2$ under fixed $\boldsymbol{U}$. However, our weighted formulation (Appendix C) accommodates dynamic/multi-scale $\boldsymbol{U}$ where frequency importance varies. We will clarify this in the revision.
>
> > **Q5. Was a spectral-only ablation performed?**
>
> While we did not perform a spectral-only ablation in this work, exploring a dedicated spectral-only generative framework is an active work in progress.
>
> > **Q6. Why not use linear objective $\mathscr{L}\_p + \mathbb{E}[\|\boldsymbol{U}\boldsymbol{x}\_0 - f(\boldsymbol{x}\_t)\|^2]$? Is the complex CF necessary for 1-step?**
>
> Regressing the linear projection $\boldsymbol{U}\boldsymbol{x}\_0$ is equivalent to standard $\hat{\boldsymbol{x}}\_0$-prediction, forcing the network to output the conditional mean $\mathbb{E}[\boldsymbol{x}\_0 | \boldsymbol{x}\_t]$. In 1-step generation, this causes "regression to the mean" and blurry outputs.
>
> Conversely, the complex exponential mapping $\boldsymbol{x}\_0 \mapsto e^{i\boldsymbol{U}\boldsymbol{x}\_0}$ is non-linear. The conditional expectation $\mathbb{E}[e^{i\boldsymbol{U}\boldsymbol{x}\_0} | \boldsymbol{x}\_t]$ encapsulates the entire distribution, preserving high-order moments and global topology. Extracting the phase from this predicted CF bypasses the blurring effect of linear regression.
>
> > **Q7. How is conditional generation handled in PiLI? Is a CFG-style combination used?**
>
> Following Section 4.4, the class label $y$ is embedded as a global context token, allowing the network to learn the conditional CF $\phi\_{\boldsymbol{x}\_0 | y}$.
>
> During 1-NFE inference, we predict the conditional phase $\boldsymbol{\theta}\_y = \arg(\hat{\boldsymbol{\phi}}\_{\boldsymbol{x}\_0}(\boldsymbol{U}; \boldsymbol{x}\_t, t, y))$ and feed it into the PiLI analytical inversion. This approach achieves conditional generation while avoiding the double-evaluation cost of CFG. We have clarified this in the revision.

---

> > ### Author Rebuttal · Reviewer_Xqnw · 2026-04-03
> >
> > I appreciate the authors responding to some of the issues raised in the review. There are still several unaddressed or unsatisfactorily addressed concerns.
> >
> > 1/ The issue about memorization has not been addressed at all. This is important to address given the very good FIDs obtained by the proposed scheme. Lack of this evidence is a key weakness as mentioned in the Weaknesses section.
> >
> > 2/ The argument that $\mathscr{A} \propto \mathscr{B}$ is invalid since the integrals in the expressions are over $\mathbb{R}^d$. If the estimates are confined to $u$ having a constant norm, then, it does not correspond to the domain specified in the integrals and hence not an appropriate estimate. The authors have not explained how good their estimates are and in what sense.
> >
> > 3/ Regarding the alternative loss $\mathscr{L}\_{p} + \mathbb{E}[\| Ux\_{0} - f(x\_{t}) \|^{2}]$, the authors argue that such a formulation would lead to “regression to the mean” and consequently blurry outputs. However, the proposed approach also involves predicting $\mathbb{E}[e^{iUx_{0}} \mid x_{t}]$ which is still a conditional expectation, in a complex-valued space. In this case also there is averaging. The authors have not clarified why this averaging would be better than "regression to the mean"? One ultimately estimates $x_{0}$ from $e^{iUx\_{0}}$.
> >
> > 4/ Estimating higher-order moments is unreliable than estimating the mean. Therefore, it is not clear how having $e^{iUx\_{0}}$ in the formulation dramatically improves the overall generation performance.
> >
> > 5/ While the FIDs are impressive, there is no convincing justification as to what is leading to such impressive FIDs.

---

> > > ### Author Response · Authors · 2026-04-04
> > >
> > > We thank the reviewer for their feedback and hope these clarifications resolve these concerns.
> > >
> > > > **R1. Memorization concerns.**
> > >
> > > We computed k-NN cosine similarities (ResNet-50 features) between generated samples of 8 classes and the full training set. Visualizations: https://anonymous.4open.science/r/k-nn-visualization-9E65/knn-visualization.png. While Top-5 mean similarities for classes 88, 985, and 992 slightly exceed 0.95, most remain well below it.
> > >
> > > > **R2. Constant norm $u$ misaligns with $\mathbb{R}^d$ integrals, invalidating $\mathscr{A} \propto \mathscr{B}$.**
> > >
> > > Practically, $\boldsymbol{U}$ can be dynamically learned or fixed. We adopt a fixed $\boldsymbol{U}$ with constant $\|\boldsymbol{u}\_k\|\_2^2=C$, ensuring minimizing $\mathscr{A}\_{empirical}$ strictly minimizes $\mathscr{B}\_{empirical}$:
> > > $$ \mathscr{A}\_{empirical} \approx \frac{1}{K} \sum\_{k=1}^K \|\boldsymbol{u}\_k\|\_2^2 |\Delta\phi(\boldsymbol{u}\_k)|^2 = C \cdot \frac{1}{K} \sum\_{k=1}^K |\Delta\phi(\boldsymbol{u}\_k)|^2 \propto \mathscr{B}\_{empirical} $$
> > >
> > > We acknowledge the gap between the continuous integral over $\mathbb{R}^d$ and constant-norm sampling. However, we consider that fixing the row norm of $\boldsymbol{U}$ to $\pi/\sqrt{d}$ is an acceptable practice in this work to circumvent the phase wrapping problem. As such, although adopting $t \sim \text{Logit-Normal}(0,1)$ is invalid for the Variational Lower Bound and Optimal Transport, which rigorously rely on an unweighted integral $t \sim \mathcal{U}(0,1)$, it has become the preferred approach in engineering applications for achieving better generative quality. We remarked this in the manuscript, exploring dynamically learned multi-scale $\boldsymbol{U}$ to align with the continuous theory.
> > >
> > > > **R3. The authors must clarify why predicting the conditional expectation $\mathbb{E}[e^{i\boldsymbol{U}\boldsymbol{x}\_0} | \boldsymbol{x}\_t]$ in a complex space effectively avoids the "regression to the mean" blurriness.**
> > >
> > > It bypasses "regression to the mean" via the distinction between the Euclidean Arithmetic Mean and directional Circular Mean. Consider a bimodal distribution: $P(\boldsymbol{x}\_0 | \boldsymbol{x}\_t) = p \delta(\boldsymbol{x}\_A) + (1-p) \delta(\boldsymbol{x}\_B)$.
> > >
> > > **Arithmetic Mean:** For the linear objective $\mathcal{L}\_p + \mathbb{E}[\|\boldsymbol{U}\boldsymbol{x}\_0 - f(\boldsymbol{x}\_t)\|^2]$, the optimal estimation $\mathcal{M}\_{\text{linear}}$ is:
> > > $$ \hat{\boldsymbol{x}}\_{\text{linear}} = \mathcal{M}\_{\text{linear}}[P] = p \boldsymbol{x}\_A + (1-p) \boldsymbol{x}\_B $$
> > > This convex combination $\hat{\boldsymbol{x}}\_{\text{linear}} \notin \{\boldsymbol{x}\_A, \boldsymbol{x}\_B\}$ applies frequency-invariant linear interpolation, manifesting as "blurriness" in high-frequency details (e.g., sharp boundaries).
> > >
> > > **Circular Mean:** Predicting $\mathbb{E}[e^{i\boldsymbol{U}\boldsymbol{x}\_0} | \boldsymbol{x}\_t]$ with PiLI's phase extraction yields $\mathcal{M}\_{\text{circular}}$. For frequency $\boldsymbol{u}$, let $\theta\_A = \boldsymbol{u}^\top \boldsymbol{x}\_A$, $\theta\_B = \boldsymbol{u}^\top \boldsymbol{x}\_B$, and $\Delta\theta = \theta\_B - \theta\_A$. The extracted phase is:
> > > $$ \hat{\theta}\_{\text{complex}} = \mathcal{M}\_{\text{circular}}[P, \boldsymbol{u}] = \theta\_A + \arg \big( p + (1-p) e^{i\Delta\theta} \big) $$
> > > This acts as a frequency-dependent non-linear filter governed by $\Delta\theta$. As $\Delta\theta \to 0$ (via $e^{i\Delta\theta} \approx 1 + i\Delta\theta$):
> > > $$ \lim\_{\Delta\theta \to 0} \hat{\theta}\_{\text{complex}} = p\theta\_A + (1-p)\theta\_B = \boldsymbol{u}^\top \hat{\boldsymbol{x}}\_{\text{linear}} $$
> > > Here, $\mathcal{M}\_{\text{circular}}$ degenerates to $\mathcal{M}\_{\text{linear}}$, ensuring stable macroscopic structures. Conversely, as $\Delta\theta \to \pi$, the mapping becomes highly non-linear ($\arg(p - (1-p)) = \arg(2p - 1)$):
> > > $$ \lim\_{\Delta\theta \to \pi} \hat{\theta}\_{\text{complex}} = \begin{cases} \theta\_A, & p > 0.5 \\\\ \theta\_B, & p < 0.5 \end{cases} $$
> > > This rejects linear interpolation for distant modes, "snapping" to the dominant one. Unlike $\mathcal{M}\_{\text{linear}}$'s indiscriminate blurring, $\mathcal{M}\_{\text{circular}}$ continuously transitions from a linear low-pass filter ($\Delta\theta \to 0$) to a non-linear $\arg\max$ selector ($\Delta\theta \to \pi$).
> > >
> > > > **R4&5. Higher-order moments are unreliable. Why does $e^{i\boldsymbol{U}\boldsymbol{x}\_0}$ yield impressive FIDs?**
> > >
> > > Our method does not estimate unbounded higher-order moments, but computes the strictly bounded CF $e^{i\boldsymbol{U}\boldsymbol{x}\_0}$, guaranteeing stability. Practically, we use Euler decomposition (Eq. 15) with complex components from iFairy. Our VAE-free experiments are trained from scratch in pixel space, where competitive FIDs are well-established for recent diffusion models. After the paper, our code will be released.

---

### Decision · Program_Chairs · 2026-04-30

**Decision:**

Accept (regular)

**Comment:**

This paper presents a novel Primal-Spectral generative framework using a Pseudoinverse Lévy Inversion solver for efficient 1-NFE generation. Reviewers praised the elegant spectral domain formulation, noting it yields state-of-the-art results at remarkably low computational costs (Reviewers mnTD, CumU, Xqnw). Initial critiques highlighted presentation gaps regarding spectral classifier-free guidance (Reviewer mnTD), omitted connections to frequency-domain GANs (Reviewer CumU), and deeper concerns regarding potential data memorization and the distinction between linear and circular means (Reviewer Xqnw). The authors' comprehensive rebuttal successfully resolved these issues by adding explicit pseudocode, contextualizing prior literature, including fairer baseline comparisons, and clarifying the non-linear mechanics of phase extraction. With these improvements, the paper’s highly efficient one-step generation capabilities and solid theoretical foundation make it a great contribution to the field.